

# Quantum Bose-Fermi droplets

**Debraj Rakshit[1], Tomasz Karpiuk[2], Mirosław Brewczyk[2*] and Mariusz Gajda[1]**

**1** Institute of Physics, Polish Academy of Sciences,
Aleja Lotników 32/46, PL-02668 Warsaw, Poland
**2** Wydział Fizyki, Uniwersytet w Białymstoku,
ul. K. Ciołkowskiego 1L, 15-245 Białystok, Poland

* m.brewczyk@uwb.edu.pl

## Abstract

We study the stability of a zero temperature mixture of attractively interacting degenerate bosons and spin-polarized fermions in the absence of confinement. We demonstrate that higher order corrections to the standard mean-field energy can lead to a formation of Bose-Fermi liquid droplets – self-bound systems in three-dimensional space. The stability analysis of the homogeneous case is supported by numerical simulations of finite systems by explicit inclusion of surface effects. We discuss the experimental feasibility of formation of quantum droplets and indicate the main obstacle – inelastic three-body collisions.

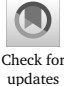

## 1 Introduction

Self-bound systems are quite common in nature. They appear at different scales. Atomic nuclei, Helium droplets, or astronomical objects like white dwarfs or neutron stars are some of

the prominent examples. Their stabilization mechanism is due to a subtle balance of attractive forces and repulsive interactions.

Yet another, not known so far self-bound systems, namely extremely dilute quantum liquid droplets of ultracold atoms, have been suggested to exist in a mixture of two Bose-Einstein condensates of different species, [1]. Soon after this prediction the quantum droplets were unexpectedly discovered in quite a different setting – in ultracold $^{164}$Dy gas, i.e. a dilute gas composed of atoms possessing the largest dipolar magnetic moment among all atomic species [2]. Further experiments followed [3–5], and soon $^{166}$Er droplets, with dipole-dipole interactions as a crucial element, were created [6]. Quite recently another self-bound objects – droplets in a two-component mixture of $^{39}$K atoms entered the stage [7–9]. These ones are a direct realization of the scenario suggested by Petrov [1].

Quantum liquid droplets having densities of about $10^{15}$cm$^{-3}$, about eight orders of magnitude less than Helium droplets, are the most dilute droplets ever. They also exist in reduced dimension space for both the dipolar case [10] as well as the mixture case [11], however the dimensional crossover region seems to be easier to access in experiments [12,13]. Droplets of ultracold atoms are stabilized against a collapse by quantum fluctuations, i.e. the energy of the Bogoliubov vacuum [1]. The stabilization mechanism of quantum droplets is universal. The beyond mean-field effects responsible for quantum fluctuations can be incorporated into the general mean-field description by including the so called Lee-Huang-Yang (LHY) term [14–16] into the standard scheme based on the Gross-Pitaevskii equation [17, 18]. This extended Gross-Pitaevskii (eGP) equation allows for a self-bound state [19,20]. In addition to the eGP approach, Monte Carlo techniques, allowing for direct treatment of the beyond mean-field effects, are being employed [21–23].

In a quasi 1D geometry quantum droplets show many similarities to bright solitons. The common feature is that in both systems a quantum spreading is suppressed. Strongly bound bright solitons in Potassium condensate have been studied recently [24]. The solitons produced in $^{39}$K have a very large peak density $\sim 5 \times 10^{14}$cm$^{-3}$, and exist, similarly as droplets, at the edge of collapse of the system. It was shown experimentally [8] that in a mixture of two spin states of $^{39}$K a transition from lower density bright solitons to quantum droplets of higher densities has the character of a first-order phase transition. In the crossover region both solitons and droplets exist.

This analogy allows to invoke yet another system supporting bright solitons. The effective interactions between bosons can change their character and become attractive in a 1D ultracold mixture of mutually repeling Bose atoms attracted to polarized Fermi atoms. Bright solitons can be expected then. This scenario was suggested in [25] where $^{40}$K and $^{87}$Rb were fermionic and bosonic agents, respectively. This choice is particularly convenient because of the large natural attraction between the two species. Only recently it was verified experimentally that for an appropriately chosen attraction between bosons and fermions, the Bose-Fermi mixture is turned into a train of Bose-Fermi solitons [26].

Effective Bose-Bose interactions become attractive only at the edge of stability of a system [27, 28]. In this paper we want to study such Bose-Fermi systems in the unstable region. The main question we want to pose is whether quantum fluctuations contributing to the energy of the Bose component and/or higher order beyond mean-field repulsive interactions between Bose and Fermi species can stabilize the mixture and lead to a formation of dilute quantum liquids in the limit of weak interactions. Although our motivation roots in elongated quasi 1D systems, we focus here on the generic 3D case having in mind the ultracold Bose-Fermi mixture of $^{133}$Cs-$^6$Li and recent experiments of the Cheng Chin group [29].

In a recently published work, Ref. [30], a repulsive short-range three-bosons interactions are added to stabilize the Bose-Fermi mixture. This mechanism was previously suggested in [31] to stabilize a dipolar condensate. Unfortunately the mechanism is rather difficult to

implement because large three-body elastic collisions are typically accompanied by large three-body losses. In addition, in [30] it is assumed that fermions are in a fully-paired superfluid state, what in fact makes the system similar to a Bose-Bose rather than to a Bose-Fermi mixture. And finally, the Ref. [30] shows that the droplets may consist of bosonic and fermionic atoms in almost equal ratio in contrary to our results indicating a significant domination of the Bose component.

## 2 Uniform mixture

The mean-field energy $E_0$ of a uniform system in a volume $V$, having $N_B = n_B V$ and $N_F = n_F V$ bosons and fermions, respectively, can be written in the form:

$$E_0/V = \varepsilon_0(n_B, n_F) = 3\,\varepsilon_F\,n_F/5 + g_B\,n_B^2/2 + g_{BF}\,n_B n_F, \tag{1}$$

where $n_B$ and $n_F$ are atomic densities, and the consecutive terms (energy densities) correspond to: (1) the kinetic energy of fermions with $\varepsilon_F = \hbar^2 k_F^2/2m_F = 5\,\kappa_k n_F^{2/3}/3$ being the Fermi energy, and $k_F = (6\pi^2 n_F)^{1/3}$ the Fermi wave number, (2) the boson-boson interaction energy, and finally (3) the boson-fermion contact interaction energy. For convenience we introduced the following notation: $\kappa_k = (3/10)(6\pi^2)^{2/3}\hbar^2/m_F$, $g_B = 4\pi\hbar^2 a_B/m_B$, and $g_{BF} = 2\pi\hbar^2 a_{BF}/\mu$, where $a_B$ ($a_{BF}$) is the scattering length corresponding to the boson-boson (boson-fermion) interactions and $m_B$, $m_F$, and $\mu = m_B m_F/(m_B + m_F)$ are the bosonic, fermionic, and reduced masses, respectively.

We assume the weak interaction limit, i.e. the gas parameters are small: $n_B^{1/3} a_B \ll 1$ and $n_F^{1/3} a_{BF} \ll 1$ so the kinetic energy of fermions, being proportional to $n_F^{5/3}$, is the largest contribution to the system energy. It favors a spreading of the fermionic component all the way to infinity. Similar is the effect of repulsive boson-boson interactions, $g_B > 0$. However, a sufficiently strong attraction, $g_{BF} < 0$, can suppress this expansion, but the equilibrium reached is unstable. Higher order terms must come into play to ensure stability. A perturbative approach suggests the Lee-Huang-Yang term (LHY), the zero-point energy of the Bogoliubov vacuum of a Bose system:

$$E_{LHY}/V = \varepsilon_{LHY}(n_B, n_F) = C_{LHY}\,n_B^{5/2}, \tag{2}$$

with $C_{LHY} = 64/(15\sqrt{\pi})\,g_B\,a_B^{3/2}$. However, this is not enough to stop the system from expansion. Our studies indicate that the contribution to the mutual boson-fermion interaction resulting from the higher order term in the Bose-Fermi coupling turns out to be the most important. These effects were considered on a theoretical ground [32–35]. A Bose-Fermi system across a broad Feshbach resonance was studied in [34], but results obtained are applicable only when the fermionic density is much larger than the density of bosons. As it will be shown later, this situation does not support droplets formation. The contribution to the Bose-Fermi interaction energy obtained in a frame of second-order perturbation theory in [33], more general than the results of [32] based on renormalized T-matrix expansion, leads to the following quantum correction to the energy:

$$\begin{aligned} E_{BF}/V &= \varepsilon_{BF}(n_B, n_F) = \varepsilon_F n_B (n_F a_{BF}^3)^{2/3} A(w, \alpha) \\ &= C_{BF}\,n_B n_F^{4/3} A(w, \alpha), \end{aligned} \tag{3}$$

where $w = m_B/m_F$ and $\alpha = 2w(g_B n_B/\varepsilon_F)$ are the dimensionless parameters, $C_{BF} = (6\pi^2)^{2/3}\hbar^2 a_{BF}^2/2m_F$, and the function $A(w, \alpha)$ is given in a form of integral [33]

$$A(w, \alpha) = \frac{2(1+w)}{3w} \left(\frac{6}{\pi}\right)^{2/3} \int_0^\infty dk \int_{-1}^{+1} d\Omega$$
$$\left[1 - \frac{3k^2(1+w)}{\sqrt{k^2+\alpha}} \int_0^1 dq q^2 \frac{1 - \Theta(1 - \sqrt{q^2+k^2+2kq\Omega})}{\sqrt{k^2+\alpha} + wk + 2qw\Omega}\right],$$
$$(4)$$

where $\Theta(x)$ is the step theta-function. The arguments of $A(w, \alpha)$ are: $w = m_B/m_F$ and $\alpha = 2w(g_B n_B/\varepsilon_F) = 16\pi n_B a_B^3/(6\pi^2 n_F a_B^3)^{2/3}$. The above formula, Eq. (3), coincides with the results of [32] for $\alpha \ll 1$, i.e. in the limit when the Fermi energy is much larger than the chemical potential of bosons (assuming the mass ratio, $w$, is of the order of one).

Summarizing the above discussion, in the regime where both gas parameters are small, $a_B n_B^{1/3} \ll 1$ and $a_{BF} n_F^{1/3} \ll 1$, we approximate the energy of the dilute uniform system by the following expression:

$$E(N_B, N_F, V) = E_0 + E_{LHY} + E_{BF} = V \varepsilon(n_B, n_F). \qquad (5)$$

To find densities corresponding to the energy minimum, physical constrains should be introduced. For a trapped system they are set by the number of atoms in every component. Here the system is free and we look for a configuration which is stable in absence of any external confinement. Therefore neither the volume $V$, nor the number of atoms $N_B$ and $N_F$, are controlled. Instead, the pressure $p(n_B, n_F) = -dE/dV$ plays an important role. Outside of the droplet it is equal to zero. The same must be inside. We ignore for a moment surface tension and consider an infinite system. For finite droplet, the energy related to the surface is proportional to its area $\propto V^{2/3}$, thus is only a fraction $\propto V^{-1/3}$ of the internal energy. This fraction vanishes in the limit of infinite system, reducing the importance of the surface effects. This fact will be demonstrated in the next section. With all limitations mentioned above, the condition of vanishing pressure is necessary for the mechanical stability of the system:

$$p(n_B, n_F) = n_B \mu_B + n_F \mu_F - \varepsilon(n_B, n_F) = 0, \qquad (6)$$

where we introduced the chemical potentials $\mu_{B(F)} = dE/dN_{B(F)} = \partial\varepsilon/\partial n_{B(F)}$ of both species. Numerical solutions of Eq. (6) are shown graphically in Fig. 1 for several values of $a_{BF}/a_B$ and in the case of the $^{133}$Cs -$^6$Li ($w = 22.095$) mixture (contours for the $^{41}$K -$^{40}$K ($w = 1.025$) system look similar). Pressure vanishes on the closed contours forming a kind of loops in the $n_B - n_F$ plane. If $|a_{BF}|/a_B$ is smaller than some critical value, $|a_{BF}|/a_B < \eta_0$, then Eq. (6) has no solutions. Contours marked by solid lines support negative energy solutions. They shrink with increasing $|a_{BF}|/a_B$.

All points on a single contour define mechanically stable droplets for given values of interaction parameters. The volume of the droplet is not specified, it is a scale parameter and if fixed it allows to determine the number of particles. What is most important, the energy of the system (for a fixed volume) varies along the contour, and reaches the minimal value, if:

$$\mu_B \frac{\partial p}{\partial n_F} - \mu_F \frac{\partial p}{\partial n_B} = 0. \qquad (7)$$

Eq. (7) originates in a necessary condition for the extremum of the energy density $\varepsilon(n_B, n_F)$ constrained to the zero-pressure line. The energy minima are marked in Fig. 1 by dots. Only these particular spots define systems which are stable with respect to evaporation process. If the initial ratio of the number of particles of the Bose and Fermi components is different than

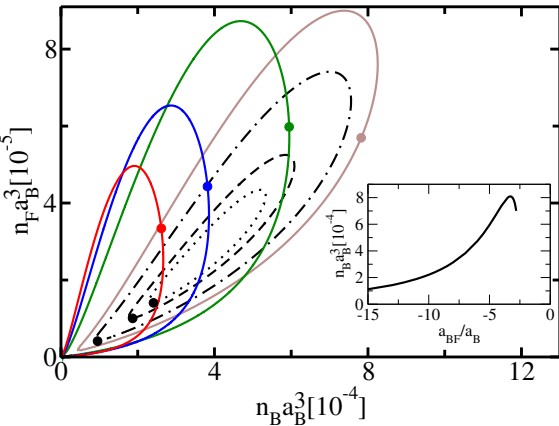

Figure 1: Solutions of Eq. (6), in the form of contour plots in the $n_B - n_F$ plane. $^{133}$Cs -$^6$Li mixture: Broken lines show metastable cases for $a_{BF}/a_B = -2.45$(dotted), $-2.5$(dashed), and $-2.7$(dash-dotted). Solid lines show stable cases for $a_{BF}/a_B = -3$(brown), $-5$(green), $-7$(blue), and $-9$(red). Dots correspond to the energy minima. The inset shows the equilibrium density of the bosonic species as a function of $a_{BF}/a_B$ for the $^{133}$Cs-$^6$Li mixture.

Table 1: The second column: Values of the critical ratio $|a_{BF}|/a_B$ supporting the existence of stable $^{41}$K -$^{40}$K, $^{87}$Rb -$^{40}$K, and $^{133}$Cs -$^6$Li mixtures. The third and the fourth column show the corresponding bosonic and fermionic densities. The fourth and fifth one give the values of the $\alpha$ parameter and the mass ratio.

| | $\eta_c$ | $n_B\, a_B^3$ | $n_F\, a_B^3$ | $\alpha$ | $m_B/m_F$ |
|---|---|---|---|---|---|
| $^{41}$K -$^{40}$K | 12.1 | $9.10 \times 10^{-6}$ | $1.26 \times 10^{-6}$ | 0.258 | 1.025 |
| $^{87}$Rb -$^{40}$K | 10.4 | $1.95 \times 10^{-5}$ | $2.01 \times 10^{-6}$ | 0.406 | 2.175 |
| $^{133}$Cs -$^6$Li | 2.8 | $7.16 \times 10^{-4}$ | $4.75 \times 10^{-5}$ | 1.796 | 22.095 |

satisfying the above mentioned condition, some particles, mostly the excess ones, are simply evaporated from the droplet.

With our choice of the energy zero, the stable self-bound droplets should be characterized by negative energy. For some range of the parameter, $\eta_0 < |a_{BF}/a_B| < \eta_c$ energies are positive. These are metastable states, marked by broken lines in Fig. 1. Only if $|a_{BF}|/a_B$ exceeds some critical value, $|a_{BF}|/a_B > \eta_c$, the energy of droplets becomes negative. These are the stable droplets. Values of $\eta_0$ and $\eta_c$ can be found numerically. In general, except of a small region of the parameter $|a_{BF}|/a_B$, the larger the attraction the smaller the equilibrium densities (see inset of Fig. 1).

In order to reach the desired ratio of $|a_{BF}|/a_B$ the two approaches are possible. One is to increase $|a_{BF}|$, the second is to tune the Bose-Bose scattering length, $a_B$, to small values. Both scenarios assume utilizing appropriate Feshbach resonances. Simultaneously one should avoid large values of densities as it would lead to a relatively large atom number decay due to three-body recombination. This is why the life-time of a droplet is typically of the order of 10 ms [7, 8]. The life-time of Bose-Fermi droplets will be estimated in the following section, where we study dynamical situations.

In the first column of Tab. 1 we list values of the critical ratio $\eta_c$ supporting stable liquid droplets, while in the second and third columns we list corresponding densities of Bose and Fermi species. We present the results for three different mixtures of different mass ratio, $^{41}$K-$^{40}$K, $^{87}$Rb-$^{40}$K, and $^{133}$Cs-$^6$Li. For all these mixtures bosons are in vast majority. There-

fore, fermions can be treated as impurities immersed in a bosonic cloud bringing analogy to a polaron.

Boson-fermion attraction mediates an effective attraction between fermionic atoms. It prevents expansion of fermions due to quantum pressure. A similar mechanism, the effective attraction between distinct electrons mediated by interaction with phonons is responsible for formation of Cooper pairs. The question of fermionic superfluidity of Bose-Fermi droplets seems to be legitimate. The same interaction induce an effective attraction between bosons. For a large enough number of bosons this might result in a collapse of the bosonic component. Then, fermions start to play an important role. They are able to counteract, due to quantum pressure, the collapse of bosonic cloud – an analogy with white dwarf and neutron stars becomes immediate. Hence, the studying of 'atomic white dwarfs' in the laboratory seems to be possible with Bose-Fermi droplets.

## 3 Finite system analysis – hydrodynamic approach

We address now the properties of finite Bose-Fermi droplets including surface effects. Evidently, additional energy terms related to the density gradients have to be considered. Therefore, we apply the local density approximation and add the kinetic energy $E_k^B = \int d\mathbf{r}\, \varepsilon_k^B$ with $\varepsilon_k^B = (\hbar^2/2m_B)(\nabla\sqrt{n_B})^2$ for the bosonic component to Eq. (5). Similarly, for fermions we add the Weizsäcker correction [36] to the kinetic energy, $E_{k,W}^F = \int d\mathbf{r}\, \varepsilon_{k,W}^F$, where $\varepsilon_{k,W}^F = \xi(\hbar^2/8m_F)(\nabla n_F)^2/n_F$, with $\xi = 1/9$ [37, 38]. We neglect the contribution due to higher order gradient terms. The total energy of a finite Bose-Fermi droplet is then given by $E[n_B(\mathbf{r}), n_F(\mathbf{r})] = \int d\mathbf{r}(\varepsilon + \varepsilon_k^B + \varepsilon_{k,W}^F)$.

The time evolution of the Bose-Fermi system can be conveniently treated within quantum hydrodynamics [39]. For that both bosonic and fermionic clouds are described as fluids characterized by density and velocity fields. Since we assume that bosons occupy a single quantum state, their evolution is governed just by the Schrödinger-like equation of motion which includes the mean-field, the LHY, and the boson-fermion interaction terms. Fermions require a special care, however. It has been already discovered many years ago that oscillations of electrons in a many-electron atom can be described by hydrodynamic equations [40]. We follow this proposal and write the hydrodynamic equations for fermions:

$$\frac{\partial}{\partial t} n_F = -\nabla(n_F\, \vec{v}_F),$$
$$m_F \frac{\partial}{\partial t} \vec{v}_F = -\nabla\left(\frac{\delta T}{\delta n_F} + \frac{m_F}{2}\vec{v}_F^2 + g_{BF}\, n_B + \frac{\delta E_{BF}}{\delta n_F}\right),$$

(8)

where $n_F(\mathbf{r}, t)$ and $\vec{v}_F(\mathbf{r}, t)$) denote the density and velocity fields of the fermionic component, respectively. $T$ is the intrinsic kinetic energy of the fermionic gas and is calculated including the lowest order gradient correction [36–38]

$$\frac{\delta T}{\delta n_F} = \frac{5}{3}\kappa_k\, n_F^{2/3} - \xi\frac{\hbar^2}{2m_F}\frac{\nabla^2\sqrt{n_F}}{\sqrt{n_F}},$$

(9)

with $\xi = 1/9$.

The convenient way to further treat Eqs. (8) is to bring them into a form of the Schrödinger-like equation by using the inverse Madelung transformation [41–43]. This is just a mathematical transformation which introduces a single complex function instead of density and velocity fields used in the hydrodynamic description. Both treatments are equivalent provided the

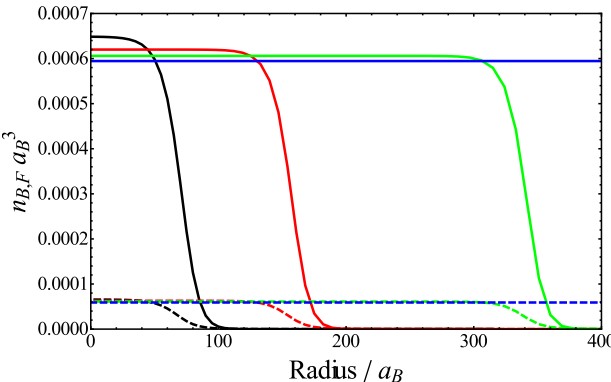

Figure 2: Radial densities (solid and dashed lines for bosons and fermions, respectively) for a sequence of Bose-Fermi droplets. for $^{133}$Cs -$^6$Li mixture for $a_{BF}/a_B = -5$ and the initial number of bosons (fermions) equal to 1000 (100), 10000 (1000), and 100000 (10000). The horizontal lines are the bosonic and fermionic densities coming from the analysis ignoring the surface effects. Clearly, the surface effects for larger droplets can be neglected.

velocity field is irrotational (vanishing vorticity). After the inverse Madelung transformation is applied, the equations of motion describing the Bose-Fermi mixture are turned into coupled Schrödinger-like equations for a condensed Bose field $\psi_B$ ($n_B = |\psi_B|^2$) and a pseudo-wavefunction for fermions $\psi_F = \sqrt{n_F} \exp(i\phi)$ ($n_F = |\psi_F|^2$ and $\vec{v}_F = (\hbar/m_F)\nabla\phi$)

$$
i\hbar \frac{\partial \psi_B}{\partial t} = \left[ -\frac{\hbar^2}{2m_B}\nabla^2 + g_B|\psi_B|^2 + \frac{5}{2}C_{LHY}|\psi_B|^3 \right.
$$
$$
\left. + g_{BF}|\psi_F|^2 + C_{BF}|\psi_F|^{8/3}A(\alpha) + C_{BF}|\psi_B|^2|\psi_F|^{8/3}\frac{\partial A}{\partial \alpha}\frac{\partial \alpha}{\partial n_B} \right]\psi_B,
$$
$$
i\hbar \frac{\partial \psi_F}{\partial t} = \left[ -\frac{\hbar^2}{2m_F}\nabla^2 + \xi'\frac{\hbar^2}{2m_F}\frac{\nabla^2|\psi_F|}{|\psi_F|} + \frac{5}{3}\kappa_k|\psi_F|^{4/3} \right.
$$
$$
\left. + g_{BF}|\psi_B|^2 + \frac{4}{3}C_{BF}|\psi_B|^2|\psi_F|^{2/3}A(\alpha) + C_{BF}|\psi_B|^2|\psi_F|^{8/3}\frac{\partial A}{\partial \alpha}\frac{\partial \alpha}{\partial n_F} \right]\psi_F. \quad (10)
$$

Here, $\xi' = 1 - \xi = 8/9$. The bosonic wave function and the fermionic pseudo-wave function are normalized as $N_{B,F} = \int d\mathbf{r}|\psi_{B,F}|^2$. We would like to emphasize that the fermionic pseudo-wave function has no direct physical meaning. Only the quantities which are the square of modulus of $\psi_F(\mathbf{r}, t)$ and the gradient of its phase can be interpreted as physical quantities. The Madelung transformation itself is supported by the Stokes' theorem. Provided that in a given region the condition $\nabla \times \vec{v}_F = 0$ is fulfilled, then the phase of the pseudo-wave function is defined as a curvilinear integral of the velocity $\vec{v}_F$.

To find the ground state of the Bose-Fermi droplet we solve Eqs. (10) using the imaginary time propagation technique [44]. The resulting ground state densities for $^{133}$Cs -$^6$Li mixture for three different numbers of bosons and fermions are shown in Fig. 2. For the smallest droplets the peak densities for both bosons and fermions are slightly higher than predicted by the analysis based on a uniform mixture. For bigger droplets the peak density approaches the uniform mixture solution as expected because the surface effects become less important. The stability of the droplets is verified by real time propagation of Eqs. (10). The conclusion is that the droplet reaches the state of minimal energy by evaporating mostly surplus atoms.

Next, we check whether Bose-Fermi droplets can be formed dynamically in a process of opening a harmonic trap where a mixture of bosonic and fermionic gases is prepared initially.

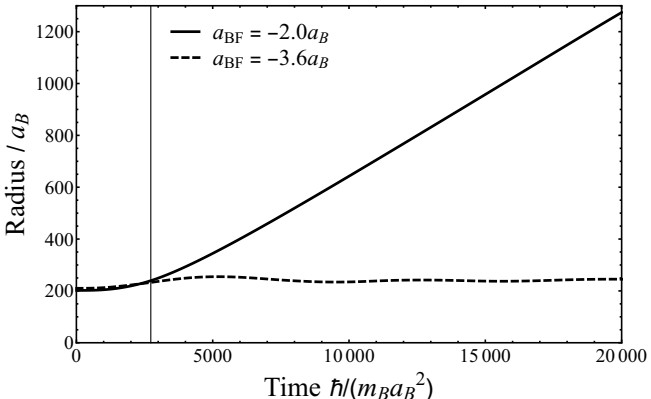

Figure 3: Width of the bosonic component of a Bose-Fermi droplet composed of $N_B = 100000$ bosons and $N_F = 10000$ fermions as a function of time. The trapping potential is removed in 1ms (marked by a vertical line). The solid (dashed) line corresponds to $a_{BF}/a_B = -2.0$ ($a_{BF}/a_B = -3.6$). For the ratio $|a_{BF}|/a_B$ equal to 2.0, which is below the critical value, both the bosonic and fermionic clouds spread out. In the case when $|a_{BF}|/a_B = 3.6$, i.e. when the attraction is larger than the critical one, a breathing droplet is formed. The width of the fermionic part of the droplet behaves similarly and is not shown.

For that we choose the $^{133}$Cs -$^6$Li mixture and set $a_B = 250\, a_0$, where $a_0$ is the Bohr radius and $a_{BF}/a_B = -2.0$ and $-3.6$. Initial numbers of atoms are: $N_B = 100000$, and $N_F = 10000$. They are confined in spherically symmetric harmonic traps with frequencies $\omega_B/(2\pi) = 200\,$Hz for bosons and $\omega_F/(2\pi) = 940\,$Hz for fermions. The trap parameters are chosen to match a radius of a droplet to be formed. We find the ground state of such a Bose-Fermi mixture by solving Eqs. (10) in imaginary time. Next, the confinement is removed in 1ms ($2730\, m_B a_B^2/\hbar$). We monitor the evolution of the system in a period of about 8 ms. The width of the bosonic component, $\int d\mathbf{r}\, r\, |\psi_B|^2/N_B$, is shown in Fig. 3. The fermionic width looks similar. The vertical line indicates the moment of time when the confinement is completely switched off. The densities preserve spherical symmetry during the evolution. Clearly, the system stays bound and a droplet is formed when the ratio $|a_{BF}|/a_B$ is above the critical value equal to about 2.8 (see dashed line in Fig. 3). Contrary, for low enough ratio $|a_{BF}|/a_B$ the bosonic and fermionic clouds spread out (solid line in Fig. 3).

The main obstacle, jeopardizing the above scenario of droplet formation is atomic loss, mainly due to three-body inelastic collisions, not included in our calculations. A crude estimation of losses can be based on the measured loss rate, $\Gamma$, of Cesium atoms from a Bose-Einstein condensate immersed in a large cloud of degenerate Fermi Lithium atoms as observed in Ref. [29].

To get a life-time for a droplet of density $n_B = 4 \times 10^{14}\,$cm$^{-3}$ corresponding to $a_B/a_0 = 250$ and $|a_{BF}|/a_B = 3.6$, we have to extrapolate the data presented in Fig. 4b of [29] assuming $a_{BF}^4$ scaling of the recombination rate $K_3$. Consistently, the loss rate scales as $\Gamma \sim a_{BF}^4\, n_B^2$. Assuming in addition a constant condensate density (equal to $5 \times 10^{13}\,$cm$^{-3}$) independent of $a_{BF}$ for all data in Fig 4b of [29] the estimated loss rate of atoms from the droplet is $\Gamma = 3000\,$s$^{-1}$. Corresponding life-time is extremely short $\tau = 0.3\,$ms. The estimation is very pessimistic and shows that Bose-Fermi droplets seem to be not feasible in present experiments. For $|a_{BF}|/a_B = 2.8$, i.e. at the edge of existence of the droplet, the life-time gets longer and reaches less pessimistic value of about $\tau = 1\,$ms.

However, the above estimation is not conclusive. Scaling of the three-body recombination

rate $K_3$ with $a_{BF}$ is actually more complex – the $a_{BF}^4$ behavior is modified by a factor which is an oscillating function of the scattering length [45]. Simple extrapolation of data presented in [29] might be not precise.

Moreover, the estimated value of the life-time is based on a particular interpretation of a stability of the system for a given densities of species in the regime of relatively large interspecies attraction $|a_{BF}|/a_0 > 520$ where a mean-field analysis predicts a collapse [29]. The stability observed in [29] is attributed to a dynamical equilibrium between losses of Li atoms trapped by a Cesium condensate and their supply from the Lithium vapor surrounding the Li-Cs system. Counter intuitively, the fast loss, mainly at the center of the cloud, stabilizes the system by preventing densities of both species to grow. A crucial assumption that densities of Cs atoms do not change with $a_{BF}$ is not confirmed by any data shown in [29].

The experiment of C. Chin's group [29] shows that a loss rate exceeds a thermalization rate in the region where the mean-field considerations predict a collapse ($|a_{BF}|/a_B = 600, 700$ in Fig. 4b of [29]). We want to speculate that even in such a dynamical situation a formation of a droplet might be still possible if both species densities adjust to the 'droplet values' at dynamical equilibrium. The situation could resemble to some extent a polariton condensate – a life-time of its components is much smaller than the coherence time of the system which is at dynamical equilibrium.

Therefore, an alternative origin of observed stability can be attributed to a repulsion of atoms due to beyond mean-field effects leading to formation of droplets of densities depending on $a_{BF}$. In such a case interpretation of Fig. 4b of [29] must be different. Accounting for dependence of densities on the scattering length will significantly influence $K_3$ scaling with $a_{BF}$. Extrapolation of loss rate is very difficult then.

To support this point we would like to note that Fig. 2b of [29] already shows an elongated falling object, living for at least by 2.5 ms, whose existence cannot be explained by the dynamical model proposed by the authors since the lack of overlap with the fermionic background cloud which was pushed upwards. We performed numerical simulations for Cs-Li mixture for parameters as studied in [29]. Our calculations show that already for $a_{BF}/a_B = -2.8$ an elongated droplet is formed in the trap with the bosonic density of $n_B = 4 \times 10^{14} \, \text{cm}^{-3}$ which after removal of the trapping potential survives and oscillates. Our simulations are in agreement with results shown in Fig. 2b, supporting Bose-Fermi droplets scenario.

The above discussion is highly speculative. Definitely much more experimental and theoretical work is needed to find out what will be the fate of the Bose-Fermi droplets discussed here. The pessimistic estimation of droplet's life-time presented above has to be treated as a serious warning but no definite conclusion about a value of loss rates in the droplet regime can be drawn on the basis of [29].

# 4 Finite system analysis – atomic-orbital approach

We address now the properties of finite Bose-Fermi droplets by using the Hartree-Fock approximation in which, as opposed to the hydrodynamic approach, fermions are treated individually. Therefore, we assign a single-particle orbital to each fermionic atom, $\psi_j^F(\mathbf{r})$, where $j = 1, ..., N_F$ and assume that the many-body wave function of the droplet is a product of the Hartree ansatz for bosons (all bosonic atoms occupy the same state $\psi_B$) and the Slater determinant, built of orbitals $\psi_j^F$, for fermions. Hence, the total fermionic and bosonic densities are $n_F = \sum_j^{N_F} |\psi_j^F|^2$ and $n_B = N_B |\psi_B|^2$, respectively. The easiest way to derive the Hartree-Fock equations of motion for the Bose-Fermi droplet is to extend our analysis performed for a uniform system (see Eq. 5) by applying the local density approximation (as we did previously) and by adding the kinetic energy density terms related to the spatial gradients

of fermionic orbitals $E_k^F = \sum_j^{N_F} \int d\mathbf{r}\, \hbar^2/(2m_F)(\nabla\psi_j^{F*})\nabla\psi_j^F$ and the bosonic wave function $E_k^B = \int d\mathbf{r}\, \hbar^2/(2m_B)(\nabla\psi_B^*)\nabla\psi_B$. The total energy of the droplet, $E + E_k^B + E_k^F$, can be now considered as a functional of bosonic wave function $\psi_B(\mathbf{r})$ and fermionic orbitals, $\psi_j^F(\mathbf{r})$. The time-dependent Hartree-Fock equations are then given by

$$
\begin{aligned}
i\hbar\frac{\partial\psi_B}{\partial t} &= \left[ -\frac{\hbar^2}{2m_B}\nabla^2 + g_B\, n_B + \frac{5}{2}C_{LHY}\, n_B^{3/2} + g_{BF}\, n_F \right. \\
&\quad + \left. C_{BF}\, n_F^{4/3}A(\alpha) + C_{BF}\, n_B n_F^{4/3}\frac{\partial A}{\partial\alpha}\frac{\partial\alpha}{\partial n_B} \right]\psi_B\,, \\
i\hbar\frac{\partial\psi_j^F}{\partial t} &= \left[ -\frac{\hbar^2}{2m_F}\nabla_j^2 + g_{BF}\, n_B + \frac{4}{3}C_{BF}\, n_B\, n_F^{1/3}A(\alpha) \right. \\
&\quad + \left. C_{BF}\, n_B\, n_F^{4/3}\frac{\partial A}{\partial\alpha}\frac{\partial\alpha}{\partial n_F} \right]\psi_j^F
\end{aligned}
\tag{11}
$$

for $j = 1, ..., N_F$.

We solve the Hartree-Fock Eqs. (11) to obtain the densities of a Bose-Fermi droplet consisting of a small number of fermions, see Fig. 4. These results are compared to the results we demonstrated in the previous section, which were achieved within the hydrodynamic description of the Bose-Fermi mixture. The atomic-orbital approach has been used by us previously to study the dynamics of Bose-Fermi solitons in quasi-one-dimensional mixtures [25, 27] (observed experimentally in [26]) as well as fermionic mixtures, in particular in the context of formation of Cooper pairs [46].

Equilibrium densities are plotted in Fig. 4, upper frame. The solutions of Eqs. (11) were obtained by adiabatic following of an initially noninteracting trapped system of $N_F$ fermions and $N_B$ bosons. Additional trapping terms in Eqs. (11) are included. The traps are chosen to match the size of the droplet to be formed. First, we gradually turn on the mutual interactions between species. In a duration of $3\times10^5\, m_B a_B^2/\hbar$ the interaction strength is changed from zero to $a_{BF} = -3a_B$. After that the harmonic trap is slowly removed within a time interval equal to $5\times10^5\, m_B a_B^2/\hbar$. Finally, we monitor the droplet during a further period of $5\times10^5\, m_B a_B^2/\hbar$. The system is stable and densities are shown in Fig. 4, upper frame. Already for as little as tens of fermions, the atomic-orbital and hydrodynamic descriptions give very similar results. On the other hand, in the lower frame of Fig. 4 we compare the dynamical properties of the Bose-Fermi mixture for different values of the mutual scattering length $a_{BF}$, obtained within the atomic-orbital and hydrodynamic approaches. Here, the trapping potential is removed in 1ms (marked by a vertical line) as in the case of Fig. 3. Similarly to the previous analysis, only for large enough $|a_{BF}|/a_B$ ($> 2.8$) a droplet is formed, otherwise we observe an expansion of both atomic clouds. Note however that for $|a_{BF}|/a_B$ close to the critical value, there appears a small discrepancy between both descriptions. But it only means that the critical values of $|a_{BF}|/a_B$ found within the atomic-orbital and hydrodynamic analyses are slightly different. This is because of relatively large contribution of the surface terms to the total energy for such small systems. These terms are treated on a different footing in both compared methods. Away of the critical value of $|a_{BF}|/a_B$ both approaches match perfectly.

The similar outcomes of hydrodynamic and Hartree-Fock calculations should not be a surprise. This is because the Madelung transformation, we invoked while solving the hydrodynamic equations in Sec. 3 can be safely used as long as the hydrodynamic velocity field is irrotational. Of course, this is not a general feature of hydrodynamic flow. For example, when vortices are present in the system, the phase of the pseudo-wave function is not defined at the positions of vortex cores and the equivalence between the Madelung and hydrodynamic approach is broken. The other example is related to the interference effect studied for fermionized bosons in the article of Girardeau *et al.* [47]. Here, two interfering clouds of fermions

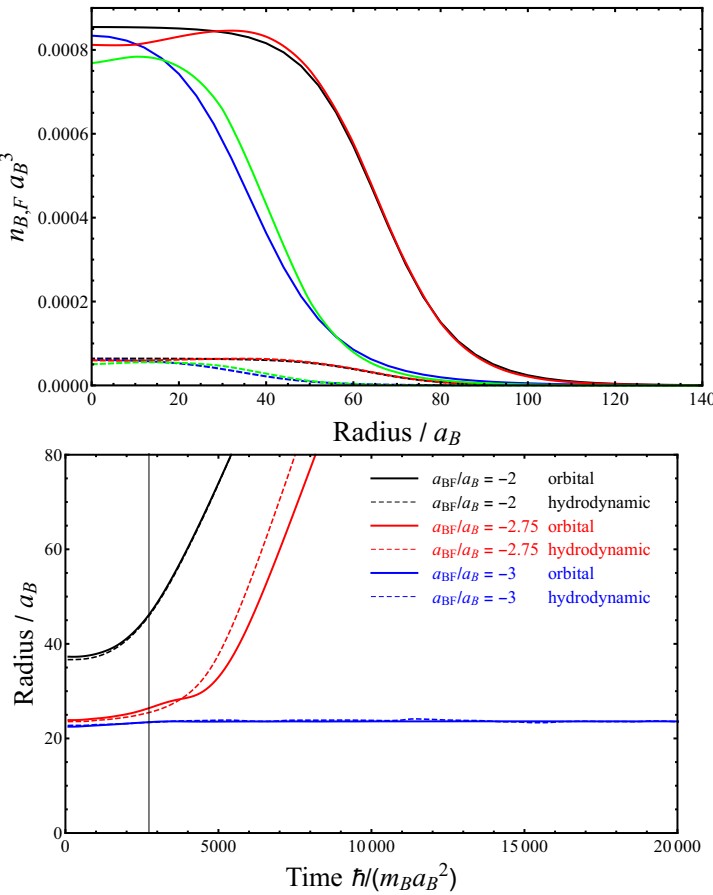

Figure 4: Upper frame: Radial densities of bosonic (solid lines) and fermionic (dashed lines) components for two Bose-Fermi droplets. Blue (hydrodynamical approach) and green (atomic-orbital method) colors correspond to a droplet consisting of 35 fermions and 350 bosons whereas colors black and red describe the case for 120 fermions and 1250 bosons. In both cases $a_{BF} = -3a_B$. Lower frame: Radius of a mixture consisting of 35 fermions and 350 bosons for different values of the coupling constants $a_{BF}$, according to the legend, as a function of time. The time is counted from the moment when the trap starts to be removed (at the time marked by the vertical line the trap is fully removed). Both stable ($|a_{BF}|/a_B > 2.8$), i.e. leading to formation of a droplet and unstable ($|a_{BF}|/a_B \leq 2.8$) cases are shown. The dashed and solid lines are obtained within the hydrodynamic and atomic-orbital approaches, respectively.

moving irrotationally can not be described as a single fluid with a potential flow. This is because the two fermionic clouds are well separated (the fermionic cloud is not simply connected) before they are merged and the total velocity (calculated as a ratio of the total current to the total density) cannot be related to the gradient of the phase in the region of vanishing density. However, in the process of adiabatic formation of the Bose-Fermi droplets the velocity field is irrotational and can be defined everywhere, therefore both hydrodynamic and Hartree-Fock descriptions give similar outcome.

Adiabatic following of the ground state of the Bose-Fermi mixture is a very effective method of finding a stationary droplet as well as occupied single particle orbitals corresponding to the Hartree-Fock Eqs. (11). The stationary version of these equations has a form of an eigenvalue problem. Solving this problem is numerically much more demanding (since it requires the

use of a large number of basis functions) but gives more insight as the fermionic orbitals and their energies are determined simultaneously. What's more, it gives not only the lowest energy orbitals occupied by a given number of fermionic atoms, but also higher energy single-particle states which could possibly be populated by fermions.

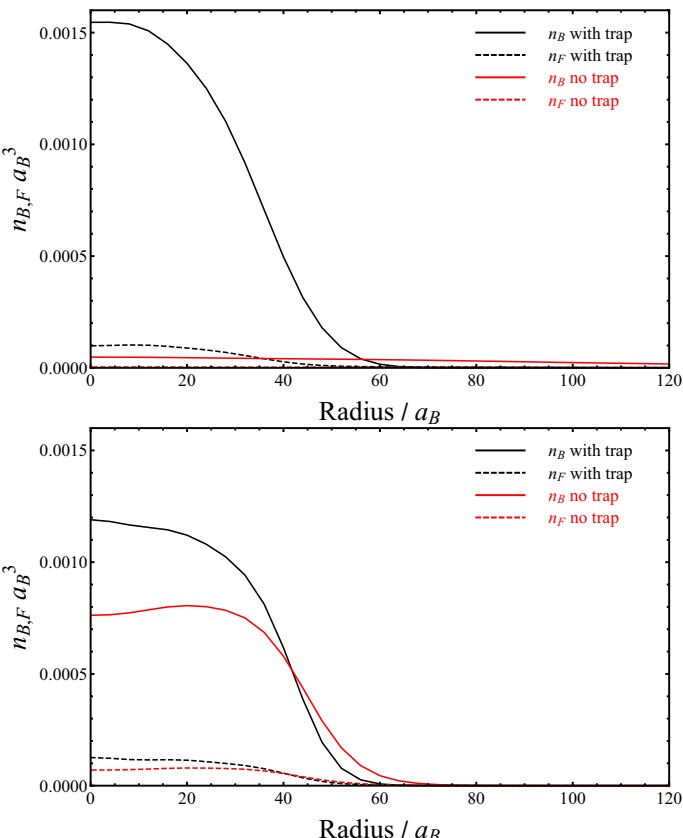

Figure 5: Bosonic densities (solid lines) and fermionic densities (dashed lines) for 35 fermions and 350 bosons – in the harmonic trap (black) and in the free space (red), for the mutual scattering length: $a_{BF}/a_B = -2$ (upper panel) and $a_{BF}/a_B = -4$ (lower panel). In the upper panel it is clearly visible that if the trap is removed the atoms are uniformly smeared over the entire grid. Fermionic density is so low that it is out of scale and is not visible in figure. Contrary, in the lower panel, after the atoms are released from the trap the densities reach equilibrium values corresponding to stationary droplets.

As previously, at the beginning we turn on the harmonic traps and decouple bosons and fermions, i.e. we put $a_{BF} = 0$. We find the density of the bosonic component by solving the first equation in Eqs. (11) by imaginary time technique. Then we solve the eigenvalue problem for the set of fermions

$$\left[ -\frac{\hbar^2}{2m_F} \nabla_j^2 + V_{FT} + g_{BF} n_B + \frac{4}{3} C_{BF} n_B n_F^{1/3} A(\alpha) + C_{BF} n_B n_F^{4/3} \frac{\partial A}{\partial \alpha} \frac{\partial \alpha}{\partial n_F} \right] \psi_j^F = \varepsilon_j^F \psi_j^F , \quad (12)$$

where $V_{FT}$ represents the harmonic trapping energy for fermions.

To this end the harmonic oscillator wave function basis is used, and the effective Hamiltonian matrix, as in Eqs. (12), is diagonalized. The eigenvectors define a new fermionic density which, in turn, allows to build the next-iteration Hamiltonian matrix. The diagonalization is then repeated and the whole cycle is done again until the fermionic orbitals energies are

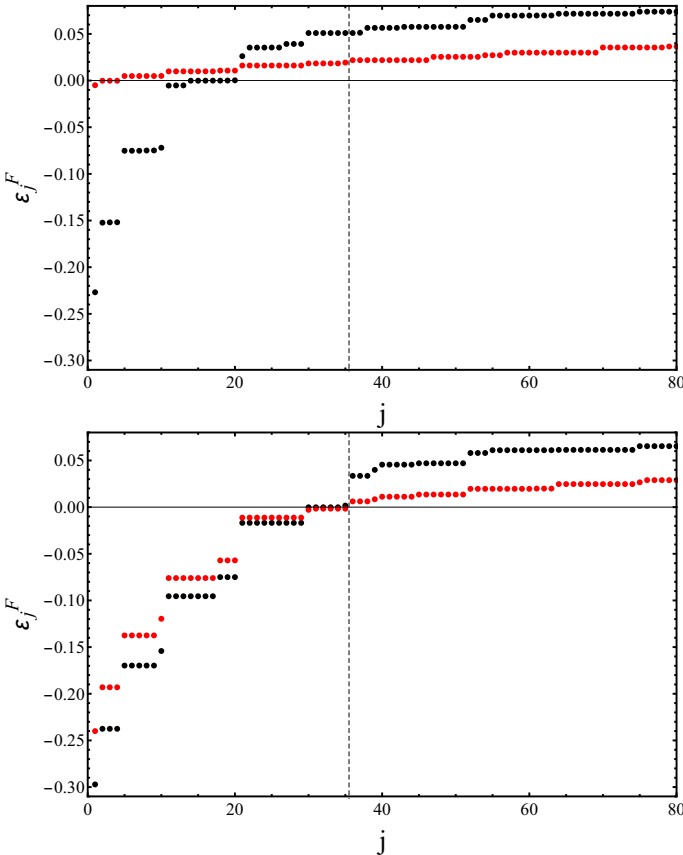

Figure 6: Single-particle energies for the system of 35 fermions immersed in a cloud of 350 bosons – in the harmonic trap (black dots), and without the trap (red dots), for the mutual scattering length equal to: $a_{BF}/a_B = -2$ (upper panel) and $a_{BF}/a_B = -4$ (lower panel). The solid horizontal line indicates zero energy level. The dashed vertical line is located at $j = 35.5$ which is exactly between 35th and 36th fermionic orbital. The total number of basis functions is 2600.

established with a sufficient accuracy. The total energy of the system is checked and provided it stabilizes within assumed error range, we move to the next step and change the scattering length $a_{BF}$ by $\Delta a_{BF} = -1a_B$ and the whole procedure is repeated. Otherwise, we do the imaginary time evolution for bosons and iterative procedure for fermions again.

Here, we report the results of such an approach for a $^{133}$Cs -$^6$Li mixture consisting of $N_B = 350$ bosons and $N_F = 35$ fermions. Atoms are confined in the isotropic harmonic trap with frequency $\omega_B/(2\pi) = 1200$ Hz for bosons and $\omega_F/(2\pi) = 4800$ Hz for fermions. The basis is formed by 3D harmonic oscillator wave functions. The trap frequency for the basis functions is $\omega/(2\pi) = 12800$ Hz. We use these basis functions since they fit better the final size of the droplets.

When the final value of $a_{BF}$ is reached we start to open the trap. We lower the trap strength by 20% in each of five steps. In Fig. 5 we show the density of bosons and fermions for two qualitatively different attractive scattering lengths. In the upper panel $|a_{BF}|/a_B = 2$, i.e. is well below the critical value 2.8. When the trap is present (black curves) both components are held by the trap. When the trap is off then both bosons and fermions spread over all available space (red lines). The red dashed line which indicates the fermionic density is so low that it is not visible in the figure. Contrary, in the lower panel of Fig. 5, when the mutual scattering length $|a_{BF}|/a_B = 4$ is well above the critical value 2.8, the behavior of the mixture is qualitatively

different. Now, when the trap is off, a stable self-bound Bose-Fermi droplet is formed.

This observation is supported by analysis of single-particle energies of the fermionic component. These are shown in Fig. 6. The upper panel corresponds to the case when $|a_{BF}|/a_B = 2$. If the system is trapped, some number of single-particle energies are negative. But when the trap is off, almost all become positive and form a continuum above zero energy. This is the case of free particles. The opposite case is when $|a_{BF}|/a_B = 4$, i.e. above the critical value 2.8. Now all 35 fermions have negative energies as one expects for trapped particles. This is indicated by the vertical dashed line located at 35.5, exactly between the 35th and 36th single-particle state. There is 8 energy levels below zero energy. One can easily recognize two families of states, corresponding to the radial quantum number equal to 0 and to 1. Because of the three-dimensional spherical geometry the eigenenergies are degenerate, and these eight energy levels are able to accommodate many more fermions than in one-dimension, where only one spin-polarized fermion per one energy level is allowed. Therefore, in three-dimensional Bose-Fermi droplets the number of fermions can be large as opposed to the case of quasi-one-dimensional Bose-Fermi mixtures of magnetic atoms [48].

## 5 Conclusions

The analysis of stability of a mixture of ultracold Bose-Fermi atoms presented here indicates that stable liquid self-bound droplets can be spontaneously formed when interspecies attraction is appropriately tuned. Droplets are stabilized by the higher order term in the Bose-Fermi coupling. We predict the values of interaction strengths as well as atomic densities corresponding to droplets of three different mixtures, suitable for experimental realization: $^{41}$K-$^{40}$K, $^{87}$Rb-$^{40}$K, and $^{133}$Cs-$^{6}$Li.

We demonstrate by time dependent calculations that a Bose-Fermi droplet should be achievable by preparing the mixture of bosonic and fermionic atoms in a trap and then by slowly removing the confinement. The main obstacle on a way to form the droplets are three-body losses. The droplets are formed in a regime where inelastic collisions are not negligible. Unfortunately, the existing experimental data does not allow to determine unambiguously the life-time of the droplets. The crude estimation based on extrapolation of the loss rate is very pessimistic, showing that Bose-Fermi droplets are illusive objects. The second scenario, assuming beyond mean-field effects play essential role, is much more optimistic. Moreover, we would like to note that low dimensional Bose-Fermi droplets [49] are free of three-body loss related troubles and should be experimentally feasible soon.

Quantum Bose-Fermi droplets bring into play the higher order term in Bose-Fermi coupling. The role of this term was not studied extensively in experiments so far. Moreover the effect of this 'correction' seems to be somewhat elusive as reported in [50]. We think that this fact should not discourage future experiments towards investigation of this higher order effects in Bose-Fermi systems. Liquid droplets such as studied here, seem to be the best systems to this end.

The ultradilute self-bound Bose-Fermi droplets are a novel, so far unknown form of matter organization. Their composition, involving not only a bosonic, but also a fermionic component, might bring into play a rich variety of physical phenomena related to polaron physics, Cooper pairing mediated by bosons, as well as fermionic superfluidity. The stabilization mechanism involving Fermi pressure brings some analogies to astronomical objects like white dwarfs or neutron stars. The dynamics of droplets, their merging and collisions can simulate some astronomical processes as well.

## Acknowledgements

The authors thank Letticia Tarruell for discussions. MG and DR acknowledge support from the EU Horizon 2020-FET QUIC 641122. MG, TK and MB from the (Polish) National Science Center Grant No. 2017/25/B/ST2/01943. Some part of the results were obtained using computers at the Computer Center of University of Białystok.

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
