# Peer review of "Quantum Bose-Fermi droplets"

_SciPost Physics, doi:SciPost Phys. 6, 079 (2019)_

## Round 3 · Referee Report · Anonymous (Referee 1) · 2018-12-18

Strengths

1- The topic is interesting and timely.

Weaknesses

1-The methods used are not valid to describe fermion dynamics.
2-The methods used are not valid to describe fermions in self-bound droplets.

Report

This manuscript theoretically investigates the possibility of self-bound Bose-Fermi droplets. It begins with an infinite uniform system analysis then moves to the finite self-bound droplet. Later, dynamic simulations are considered to test droplet stability, and droplet preparation. I do not recommend this manuscript for publication because the methods employed are inappropriate to describe the system studied.

Self-bound droplets typically support only a small number of bound states, usually a tiny percentage of the total atom number, see Petrov, PRL 115, 155302 (2015), Wenzel et al., Physica Scripta 93, 104004 (2018), and Baillie et al., PRL 119, 255302 (2017). While this is not a problem for the bosons, only one fermion can occupy each bound state. If the number of fermions exceeds the number of bound states then the excess fermions must be unbound, which will affect the stability of the entire droplet. The model in the present manuscript doesn’t account for this crucial physics, nor do they consider how the ratio of fermion number to bound state number behaves in the thermodynamic limit. Therefore, the authors cannot claim that their results are physically relevant for either the finite or uniform systems. Such considerations must be a central focus for any paper considering the self-bound nature of Bose-Fermi droplets, as was demonstrated in the paper on Bose-Fermi droplets with dipolar interactions by the Pfau group: Physica Scripta 93, 104004 (2018).

Furthermore, for the section on finite systems, the small numbers of fermions expected (typically 10s of bound states) should instead be treated as discrete modes, as was done in the paper by the Pfau group: Physica Scripta 93, 104004 (2018). Even if the fermion number was large, any continuous treatment (e.g. local density approximation) of the fermions would require a high energy cutoff (at zero energy) so that only the bound fermion states are included within the droplet.

Another significant problem is that Eq. (10) is inappropriate to describe the dynamics of fermions. Equation (10) attempts to describe all fermions within a single pseudo wavefunction. However, it is well-known that this is unphysical for describing dynamics. What would be the physical interpretation of the ‘phase’ of the fermion wavefunction? The unphysical effects on the dynamics from assigning a ‘phase’ to a single wavefunction to describe many fermions has already been clearly addressed in the literature. For example, the paper by Girardeau et al. PRL 84, 5239 (2000), which considers fermionized bosons, focusses quite a lot on this point. For another example see Kunal et al. PRA 65, 063603 (2002).

Requested changes

(see report)

  • validity: poor
  • significance: poor
  • originality: low
  • clarity: good
  • formatting: excellent
  • grammar: good

Author:  Miroslaw Brewczyk  on 2018-12-26  [id 393]

(in reply to Report 1 on 2018-12-18)

Below we shortly reply to the referee's criticism:

  1. "The methods used are not valid to describe fermion dynamics"

Our explanation must be not clear enough. Contrary to what the referee says, we do not assign a single wave function to a many-fermion system.

We use a hydrodynamic approach originated in the density functional method. The hydrodynamic equations for fermionic system can be rigorously derived based on quantum kinetic equations for reduced density matrices. A 3-component velocity field and an atomic density are used to describe the system.

If the velocity field is a gradient of a scalar potential then problem can be simplified because of reduction of the number of dynamical variables. The density (in fact its square root) and the velocity potential can be combined into a single complex function. This way the hydrodynamic equations, under some assumptions, can be mapped to a pseudo-wave function dynamics.

In a case of violent dynamics and turbulent flow one must use all components of the velocity field. This is because no potential can be assigned to the velocity field then. We do not study such a situation.

We show that the Bose-Fermi droplet can be created dynamically by slow opening of a trap. We have checked (and results might be included into the manuscript) that the atomic-orbital approach and a hydrodynamic description give very similar results already for a droplet with tens of fermions and ten times larger number of bosons. Larger systems are not tractable by this method.

  1. "The methods used are not valid to describe fermions in self-bound droplets."

In the first part we study infinite system, and we simply minimize the energy functional under physical constraints.

To account for the surface effects we modified the energy functional. Moreover in stationary case the velocity field vanishes, and our hydrodynamic description of fermions becomes the Thomas-Fermi model with the Weizsacker correction with all necessary modifications like replacing the Coulomb potential of nucleus by the effective potential produced by bosons.

The Thomas-Fermi model, used to describe multielectron atoms, has some drawbacks. It does not account for shell effects, but gives quite reasonable estimations of basic characteristics of atoms with atomic number Z. There is no better approach in the limit of infinite system than the density functional method. This is what we do for large system, being aware of limitations. We cannot do any better.

Referee correctly noticed that in a case of small systems the orbital approach is the one which is more adequate.

We have recently applied the atomic-orbital approach to a small system of Bose-Fermi mixture and confirmed the existence of Bose-Fermi droplets. These results can be incorporated into the present paper if we are given a chance.

  1. "Another significant problem is that Eq. (10) is inappropriate to describe the dynamics of fermions."

We claim that Eqs. (10) are good enough to describe adiabatic formation of the Bose-Fermi droplet. We do not claim that every dynamics can be described by Eqs. (10).

As we have already mentioned, the Madelung approach is equivalent to the hydrodynamic description if the velocity field is irrotational. However, this is not a general feature. For example, when vortices are present in the system, the phase is not differentiable at the vortices cores and the equivalence between the Madelung and hydrodynamic approaches is broken.

Finally, the paper of Girardeau et al. mentioned by the referee is related to the interference effect studied for fermionized bosons. We are well aware of this paper. Here, two interfering clouds of fermions moving irrotationally can not be, in general, described as a single fluid with a potential flow since the total velocity is calculated as a ratio of the total density current to the total density.

However, the formation of the Bose-Fermi droplets has nothing to do with creation of vortices nor with any subtleties of non-potential flow as the one studied by Girardeau et al. (the fermionic cloud is simply connected).

Author:  Debraj Rakshit  on 2019-03-02  [id 455]

(in reply to Miroslaw Brewczyk on 2018-12-26 [id 393])

We thank for the comment. Indeed the paper of S. K. Adhikari considers related, but different problem, and if we have an opportunity we shall add some critical comments in the present manuscript. We did not include the corresponding citation in the first version of our paper simply because our manuscript appeared on arXiv on 01 January 2018, well ahead of the S.
K. Adhikari's paper - sumbitted for publication much later.

Anonymous on 2019-02-26  [id 449]

(in reply to Miroslaw Brewczyk on 2018-12-26 [id 393])
Category:
remark

I am interested in the topic of this study and I find the comments by the referees to this manuscript of interest. One of the referees commented that
" This submission appears to be the first work considering self-bound Bose-Fermi mixtures stabilized by Lee-Huang-Yang effects with only contact interactions."
This statement is not to the point. I find a very similar work (S K Adhikari 2018 Laser Phys. Lett. 15 095501) demonstrating the existence and studying the properties of self-bound Bose-Fermi mixtures stabilized by Lee-Huang-Yang effects with only contact interactions using essentially the same Hamiltonian and same analytical and numerical procedures. Moreover, that paper goes beyond and considers the effect of three-body force on these states. This paper was published long before this manuscript was submitted for publication. The authors should clearly acknowledge this fact in the Abstract and Introduction of the manuscript and point out what is new or different in their study for the benefit of the reader.

---

## Round 3 · Referee Report · Anonymous (Referee 2) · 2018-12-21

Strengths

  1. The current manuscript certainly touches on a highly topical and interesting area. A feasible approach to dilute quantum droplets in free space was presented in just 2015 (for Bose-Bose mixtures [1]). Dilute dipolar gas droplets were observed in 2016 [2], and droplets of Bose-Bose mixtures very recently [3-4].

  2. This submission appears to be the first work considering self-bound Bose-Fermi mixtures stabilized by Lee-Huang-Yang effects with only contact interactions.

  3. The submission is clearly laid out.

Weaknesses

  1. The use of a single complex function to represent all of the fermions in this system is not physically reasonable. There are significant issues both for the ground state and for dynamics.

Report

  1. Ground state

In a dilute self-bound system there do not tend to be many bound states. For example Fig. 3(a) of [5] identifies in the order of 10 modes for fermionic impurities in a dipolar Bose droplet, and [1] identifies only a few modes for a Bose-Bose mixture. At most one fermion can occupy each bound state. Any additional fermions in addition to those in bound modes will occupy the continuum and escape from the mixture. Since the number of bound modes is small, each of them should be treated discretely, as [5] did for fermions trapped in a dipolar Bose droplet.

There will be a balance. The system needs to be deeply self-bound to have enough discrete modes below the energy cutoff at the continuum, but not so deeply bound that the density is too high. The loss from three-body interactions would then make the time-scales unfeasible in experiment.

  1. Dynamics

The phase of a single Fermi wavefunction cannot generally be used to represent the dynamics of the fermions, as shown by Ref. [6] (with mapping from bosons).

The submission cites three papers related to the single Fermi wavefunction, two on very different systems (metal clusters and a single helium atom) and Ref. [7]. The only comparison to experiment or other methods in Ref. [7] seems to be Fig. 1. The difference of the spin dipole mode from one (the non-interacting limit) is either small or the difference from one is comparable to the difference from experiment.

[1] D. S. Petrov, Phys. Rev. Lett. 115, 155302 (2015) [2] M. Schmitt et al, Nature 539, 259 (2016) [3] C. R. Cabrera et al, Science 359, 301 (2018) [4] G. Semeghini et al, Phys. Rev. Lett. 120, 235301 (2018) [5] M. Wenzel et al, Phys. Scr. 93 104004 (2018) [6] K. K. Das, G. J. Lapeyre, and E. M. Wright, Phys. Rev. A 65, 063603 (2002) [7] P. T. Grochowski et al, Phys. Rev. Lett. 119, 215303 (2017)

Requested changes

  1. Use a methodology that allows for the small number of discrete modes available to the fermions.

  2. With that methodology show the bounds on the number of bosons and fermions for a droplet to exist and how that varies with interaction strength.

  3. For reasonable interaction strengths, then give corresponding peak densities in SI units to allow comment on three-body loss.

  • validity: low
  • significance: good
  • originality: ok
  • clarity: good
  • formatting: good
  • grammar: reasonable

Author:  Miroslaw Brewczyk  on 2019-01-17  [id 406]

(in reply to Report 2 on 2018-12-21)
Category:
reply to objection

We performed further calculations by using the Hartree-Fock method (in reply to the requested change number 1) and in the attached file we compare the results with the ones included already in the main text, obtained within the hydrodynamic approach.

Attachment:

comment.pdf

Matthew Davis  on 2019-02-05  [id 427]

(in reply to Miroslaw Brewczyk on 2019-01-17 [id 406])
Category:
suggestion for further work

Following on from your additional calculations, below I relay further specific comments from one of the referees of your initial submission. I will hope this will help expedite the next round of refereeing:

  1. As the self-bound droplet is not trapped, the effective potential is finite and the number of bound states is limited before the continuum is reached. Can you please calculate the number of bound states of the bosons (i.e. the states available to the fermions)? A full Bogoliubov-de Gennes calculation would be time consuming, but a possible simple and approximate approach would be to take the effective potential of the bosons in the self-bound Bose+Fermi droplet from your calculations and find the number of non-interacting modes supported by that numerical effective potential.

  2. Can you please perform the same calculation as in your reply of 17 Jan, but applied to the solid curve in Fig. 3 of your manuscript, where there is expansion of the gas. A smaller atom number would be OK.

Author:  Mariusz Gajda  on 2018-12-28  [id 394]

(in reply to Report 2 on 2018-12-21)

The referee admits that our manuscript touches on a highly topical and interesting area, but the use of a single complex function to represent all of the fermions in this system is not physically reasonable. There are significant issues both for the ground state and for dynamics.

We agree that our approach might have weaknesses in a case of small systems of tens fermions as studied in [1]. Discreetness of fermionic states might be an issue then. We have done orbital calculations in such a situation recently. The differences are not big, and we are now able to revise the manuscript in this direction. However, orbital calculations are numerically very demanding, thus must be limited to small droplets only.

But as shown in [2] not only few fermions, but also about 400 fermionic atoms can be trapped in a bosonic cloud. Then the mean field approach is applicable.

The mean field approach used by us and rigorously transformed to the pseudo wave-function formalism (with some limitations on the velocity field) corresponds to the limit of a large number of particles. Our goal was to compare results of the first part of our manuscript, where we consider infinite system, to the results which account for the surface effects. We choose big systems then.

In a stationary case, the velocity field is zero, and there is no problem in transforming the hydrodynamic description into the pseudo wave-function formalism. As the dynamic situation is concerned, we study adiabatic opening of trap. As can be checked a posteriori, no vorticity is created then. Hydrodynamic approach and pseudo wave-function formalisms are equivalent.

In our description the fermion contribution to the mean field energy is given by the Thomas-Fermi functional with the Weizscaker term. The functional used is very similar to the one introduced by Thomas and Fermi. It is well known that binding energies of multielectron atoms given by the Thomas-Fermi model differ from the `exact ones’ by about 10%. Differences are the largest for atoms with closed shells, and become smaller with increasing atomic number. Thus, by analogy, we expect that our approach is quite accurate for larger system such as we study in the manuscript. Orbital based calculations are beyond our reach for these number of atoms.

[1] Matthias Wenzel, Tilman Pfau and Igor Ferrier-Barbut, Physica Scripta 93, 104004 (2018)
[1] B. J. DeSalvo, Krutik Patel, Jacob Johansen, and Cheng Chin, PRL 119, 233401 (2017)

---

## Round 4 · Referee Report · Anonymous · 2019-3-21

Strengths
-This a timely and relevant study on Bose-Fermi self-bound droplets.
-This manuscript now considers the discrete and finite number of bound fermion excited states.
Weaknesses
For part of their work, the authors employ a single pseudo-wavefunction to describe all fermions of the system. The ‘phase’ of this fermion pseudo-wavefunction is then used to describe [via Eqs. (10)] the non-adiabatic dynamics of an expanding droplet in Fig. 3. The authors do not adequately justify the use of this apparently unphysical term in (10). However, I do recognize that in this figure the most important result is probably the question of whether the droplet is stable or unstable [which might be better described by Eqs. (10)], rather than unstable dynamics itself.
Report
There is still one issue that remains. In Eqs. (10) the authors use a single pseudo-wavefunction to describe all fermions of a system, which they use for non-adiabatic dynamic simulations in Fig. 3. In this equation there is a term which involves the ‘phase’ of this fermion pseudo-wavefunction. The authors later claim
“… the hydrodynamic equations in Sec. 3 can be safely used as long as the hydrodynamic velocity field is irrotational.”. However, this claim seems overly strong, is unsubstantiated, and I can see no situation where such a pseudo-wavefunction ‘phase’ should describe many fermions in an actual physical dynamic process. Even after I mentioned this last time, the authors have not been able to explain this. However, I do recognize that the dynamics of the fermion pseudo-wavefunction is not a central part of this manuscript. Even for the problematic figure 3, which presents results for non-adiabatically expanding droplet, the most interesting part (in my opinion) is the instability itself, not the post-instability dynamics. For these reasons I do not request any further changes to the calculations or simulations themselves, as I do not wish to delay publication, but the authors should better address these concerns in their text. The authors should either admit that the term in Eqs. (10) which involves the ‘phase’ of the fermion pseudo-wavefunction is unphysical when used to describe many fermions, or they should explain what physical process it actually represents, and give evidence for this.
Requested changes
Text change only: The authors should either admit that the term in Eqs. (10) which involves the ‘phase’ of the single fermion pseudo-wavefunction (which they use to describe the non-adiabatic dynamics of many fermions) is unphysical, or they should explain what physical process it actually represents, and give evidence for this.
Matthew Davis on 2019-04-01 [id 478]
This opening paragraph was left out of report 1 in error by the author, and communicated to the editor separately:
"The manuscript now addresses the crucial fact that there are only a finite number of bound excitations within self-bound droplets by employing a kind of Hartree-Fock theory. Clearly this is an important effect since only one fermion can occupy each bound state. While many unanswered questions remain, I believe that these can be left for future work and I am now willing to recommend this manuscript for publication as long as the authors address my remaining concern below, which only regards a text change."

---

## Round 4 · Referee Report · Anonymous · 2019-3-25

Strengths
1. The work is still very topical.
2. The calculations are generally well justified, with two methods being compared.
Weaknesses
1. Comparison of expansion dynamics using the two computational methods is not possible due to different assumptions.
2. Information of three body loss is overly precise given uncertainties and not clearly laid out.
Report
I thank the authors for their resubmission. The work in the new Section 4 using fermionic orbitals provides very helpful justification of the earlier hydrodynamic calculations.
I recommend the manuscript is published, after quickly addressing two issues.
Requested changes
1. Dynamics with one Fermi wavefunction
I am still concerned that the solid curve in Fig. 3(left) may not be approximately correct, as it uses a single wavefunction with a single phase for an expanding gas. The behaviour of the solid line Fig. 3(left) cannot be easily compared to the black curve in Fig. 5(right) as the quench, timescales and number of atoms are quite different. In Fig. 5(right), please include dynamical results from the hydrodynamical approach with the same parameters as for the atomic-orbital approach. This will allow easy comparison, as we have for equilibrium results in Fig. 5(left).
2. Three body loss
The calculation for Fig. 4 is too approximate to be useful. The quench of scattering lengths and trap in an experiment will lead to oscillations which will affect loss, there being a trade-off between a fast quench (desirable due to short lifetimes) but stronger oscillations. The experiment will also have noise which has not been added in the calculations. I suggest removing Fig. 4 and coming up with a broad estimate based on the rate coefficient.
Also the paragraph on three body loss is confusing. Please state clearly what you get from where. You are considering a case of $a_B=250a_0$ and $a_{BF}=-3.6a_B=-900a_0$? The bosonic density of $n_B = 0.0009/a_B^3 = 3 \times 10^{14}\mathrm{cm}^{-3}$ is taken from your calculations (the manuscript says 'From the rate equation')? I see you have used $K_3=\Gamma/n_B^2$, but where did your $\Gamma=10/\mathrm{s}$ come from? Please do not state $K_3$ to three significant figures. Please state clearly how you get $\Gamma=50/\mathrm{s}$ from Fig. 4(b) of [29] including how you allow for your values of $a_B$ and $a_{BF}$ and for your increased density.
Please also discuss how 'The loss rate exceeds the thermalization rate at $a_{BF}=-520a_0$, above which the system no longer reaches thermal equilibrium' [29] relates to your system.

---

## Round 4 · Referee Report · Anonymous · 2019-3-29

Strengths
1. The work is very timely. Currently, self-bound quantum droplets represent
one the most exciting topics in the field of ultracold atoms.
2. The results are interesting and they could inspire new experiments, and also further theoretical investigation.
Weaknesses
Some points need to be clarified (see the requested changes).
Report
I would be pleased to recommend the publication of this manuscript after the authors have addressed the points mentioned below.
Requested changes
1) The author state in the abstract that "Bose-Fermi liquid droplets – self-bound incompressible systems" are formed,
but then the (in)compressibility of the system is never discussed. The authors should clarify this point, are these systems
really incompressible?
1) The choice of the time unit -- ħ/(mBaB2) -- is very uncommon. Why not using milliseconds or something more
immediate to read? I would recommend to change it or explain clearly in the text the reason of this choice.
2) The content of Fig. 4 does not justify the need of a figure. Its meaning could be easily explained by adding a text line.
3) Which is the physical origin of the oscillations in Fig. 4right?
Have the authors checked that this is not just a numerical effect?
Please explain in the text.
4) The English may need some revision. Please check carefully the (missing) articles.

---

## Round 4 · Author Response

We thank both referees for their comments on the manuscript. Both referees agree that the topic we consider is timely and interesting. Unfortunately they express worries that our approach is not valid because of applying a mean field-approach and pseudo-wave-function formalism which, as they suspect, cannot account for a correct treatment of fermionic systems in a droplet.
The Referee 1 points two major defects of the manuscript:
1-The methods used are not valid to describe fermion dynamics. 2-The methods used are not valid to describe fermions in self-bound droplets.
Referee’s 2 criticism is based upon the same arguments. The referee says:
- The use of a single complex function to represent all of the fermions in this system is not physically reasonable. There are significant issues both for the ground state and for dynamics.
The arguments on the necessity of accounting for a discrete spectrum of bound fermions are given in the two reports. In addition the referees point out to problems of the unphysical phase of the pseudo-wavefunction for fermions.
The Referee 2 requests the following changes:
- Use a methodology that allows for the small number of discrete modes available to the fermions.
- With that methodology show the bounds on the number of bosons and fermions for a droplet to exist and how that varies with interaction strength.
- For reasonable interaction strengths, then give corresponding peak densities in SI units to allow comment on three-body loss.
In the response below we show that the above mentioned worries are not justified and we substantiate our results by additional arguments and results based on numerical calculations. Because both reports address similar issues we will write a single answer where we carefully address all points raised by the two referees.
Resubmitted version of our manuscript is substantially modified. In particular:
-
We added a new section no. 4, entitled: Finite system analysis – atomic-orbital approach, where we show results based on the Hartree-Fock formalism for fermions which, contrary to the mean field description, accounts for a discreet character of the fermionic spectrum a. In the first part we present dynamical calculations for the ground state with a discreet single-particle basis for fermionic atoms (the size of the basis equals to the number of fermions). The base functions are modified adiabatically, starting from non-interacting trap system evolving towards interacting one without any trapping potential eventually. We consider both bound and unbound states. New figure 5 is added. b. In the second part we present results of extensive numerical diagonalization of the system Hamiltonian in basis of 2600 oscillatory functions. We investigate two cases: unbound and self-bound systems. In both cases we show single-particle spectrum of fermionic subsystem. Figures 6 and 7 are added.
-
We added discussion of losses and new Fig.4 illustrating how number of bosons in droplets decreases due to the three-body collisions. We give physical values of parameters for the Cesium-Lithium mixture as used in the Chin Cheng’s group experiment [1].
We did not included in the resubmitted manuscript the request #2 of the second Referee: `With that methodology show the bounds on the number of bosons and fermions for a droplet to exist and how that varies with interaction strength.’ We have to admit that this is indeed a very interesting problem. We have shown that droplets exist even for as little as 35 fermions and 350 bosons. However systematic studies using the Hartree-Fock method require extensive, time consuming numerical work. In our opinion comprehensive answer to the referee’s question deserves a separate publication.
Below we present more elaborate discussion in reply to the referees’ concerns.
- Comment on the referees’ criticism of using the pseudo-wavefunction formalism and on the mean-field description of fermionic systems.
We want to stress that the paper can be divided into two-parts: a semi-analytic (to some extend) approach where we specify conditions for equilibrium of a self-bound system with a free surface. This method, valid in the limit of infinite system, is general, and contrary to the approximate approach of D. Petrov does not base upon diagonalization of a quadratic form. It looks as if the referees are ignoring this part. In our opinion this part is very important. Results of this part were supported by numerical calculations based on the hydrodynamic approach.
In a stationary situation, both infinite system and finite system case, our approach is simply the standard Thomas-Fermi method with the Weizsacker correction included.
Evidently, the Thomas-Fermi model, introduced to describe electronic cloud and binding energies of multielectron atoms is far from the accuracy of sophisticated quantum-chemist approaches to multielectron systems. We do not claim that we are such accurate. But the TF approach in not totally wrong. It gives quite reasonable estimation (with 10% accuracy) of the binding energies of atoms, quite small systems though. Our predictions prove that the quantum fluctuations are able to stabilize the Bose-Fermi mixtures and can lead to formation of liquid droplets. We are convinced that in a case of large systems, the approach used by us is quite accurate, and we are not aware of any better than mean-field approach for systems having about 1000, 10000, or 100000 fermions or more, as we show in Fig. 2. No discreet treatment is possible for the system this big!
The static approach is generalized then to a dynamical situation by introducing a velocity field. The corresponding hydrodynamic equations are brought to the form of the Schoedinger-like equation for the pseudo-wavefunction which results from a kind of “complexification” of the density and the velocity “potential”. The transformation assumes that the velocity field is irrotational, or more precisely that it is defined on a simply connected support, and a velocity potential – a phase, can be used instead. We explain the issue of complex pseudo-wavefunction for fermions in details in the comments to the reports and we cannot add anything substantial to this discussion without repeating the same arguments. So we believe that our approach is correct as long as Thomas-Fermi model is justified.
- Elaborated discussion of the arguments against our approach bringing the controversy on continuous versus discreet approach for many fermion system.
The referees say, that self-bound Bose-Bose droplets have only few, or none, bound excited states. It’s true, all bosons can occupy the same state so one bound singe-particle state can support Bose-Bose-droplets, this is not enough in the case of fermionic component because of the Pauli principle. For fermions number of bound singe particle states gives an estimation of number of fermions which can be “trapped” in the droplet.
The referees, based on observation of [2] claim that no more than 10 one-particle states can be trapped in bosonic cloud, thus discreetness of the fermionic spectrum is crucial for bound systems. The referee 2 gives also example of Bose-Bose droplets, which have only few excited states.
The referee’s observation is correct. Number of bound states in effective potential formed by bosonic atoms must not be smaller than the number of fermions. We want to stress that we deal here with 3D situation, contrary to the effective 1D case of [2]. Note that in 3D the energy states are highly degenerate, every angular momentum state L, is (2L+1)-fold degenerate. Our approach, leading to the mean-field energy of fermionic component is based on estimation of the number of bound states in a uniform potential. And this estimation is correct up to the leading order in the Fermi energy. For a spherically symmetric harmonic oscillator, the number of bound states of energies not larger them $m \hbar \omega$ grows as $m^3$. The TF model recovers the same scaling.
Because this issue seems to be controversial, what is expressed also in the comment of Mathew Davis, in the corrected version of the manuscript we included the subsection entitled “Finite system analysis – atomic orbital approach” devoted exclusively to justification of the mean field method. We support the hydrodynamic results by results obtained in the Hartree-Fock method accounting for discreetness of fermionic orbitals.
In the first part of this section we defined the Hartree-Fock formalism equivalent to the energy functional used by us. Then we find densities of droplet by adiabatic following of the ground state of 35 fermions and 350 bosons starting from decoupled system and gradually increasing the mutual coupling. We used the basis of 35 fermionic states, which were dynamically modified according to the Hartree-Fock equations coupled to the extended Gross-Pitaevskii equation. Resulting densities of droplets agree very well with those obtained by the pseudo-wavefunction formalism. In addition we showed that for too weak Bose-Fermi attraction the system is not bound and its radius spreads in time after releasing from the trap.
In the second part we used a huge basis of oscillatory wave-functions to find fermionic single-particle states in the effective bosonic potential for a small system of 35 fermions and 350 bosons. We want to stress that these are several-month-lasting calculations. Within this approach we show that number of states bound by bosonic cloud is exactly equal to the number of fermions.
We believe that this extensive and time-consuming calculations are convincing for the two referees, moreover they justify usage of the mean field approach as well as the pseudo-wave-function formalism.
Finally, as the referee 2 requested, we assumed realistic values of densities of Cesium-Lithium systems [1], for which we calculated the peak atomic densities, estimated the loss rate and showed the results of numerical simulations of droplet dynamics with losses included. We showed that the lifetime of droplet is sufficiently long there. To illustrate this analysis we included new figure.
We think that we gave answered to all the concerns of the both referees. We modified the manuscript to account for all their criticisms and we hope that our manuscript, in the present form, will be accepted for publication.
References [1] B.J. DeSalvo et al. Phys. Rev. Lett. 119, 233401 (2017) [2] M. Wentzel et al. Physica Scripta 93, 104004 (2018)

---

## Round 4 · List of Changes

1. We added a new section no. 4, entitled: Finite system analysis – atomic-orbital approach, where we show results based on the Hartree-Fock formalism for fermions which, contrary to the mean field description, accounts for a discreet character of the fermionic spectrum
a. In the first part we present dynamical calculations for the ground state with a discreet single-particle basis for fermionic atoms (the size of the basis equals to the number of fermions). The base functions are modified adiabatically, starting from non-interacting trap system evolving towards interacting one without any trapping potential eventually. We consider both bound and unbound states. New figure 5 is added.
b. In the second part we present results of extensive numerical diagonalization of the system Hamiltonian in basis of 2600 oscillatory functions. We investigate two cases: unbound and self-bound systems. In both cases we show single-particle spectrum of fermionic subsystem. Figures 6 and 7 are added.
2. We added discussion of losses and new Fig.4 illustrating how number of bosons in droplets decreases due to the three-body collisions. We give physical values of parameters for the Cesium-Lithium mixture as used in the Chin Cheng’s group experiment.

---

## Round 5 · Referee Report · Anonymous · 2019-5-2

Report

The authors have not adequately addressed my remaining concern. In the last round of refereeing this was summarized by the request:

“The authors should either admit that the term in Eqs. (10) which involves the ‘phase’ of the single fermion pseudo-wave function (which they use to describe the non-adiabatic dynamics of many fermions) is unphysical, or they should explain what physical process it actually represents, and give evidence for this.”

In reply, the authors have added the text:

“We would like to emphasize that the fermionic pseudo-wave function has no direct physical meaning. Only the quantities which are the square of modulus of ψ_F(r,t) and the gradient of its phase can be interpreted as physical quantities. The Madelung transformation itself is supported by the Stokes' theorem. Provided that in a given region the condition ∇×v_F=0 is fulfilled, then the phase of the pseudo-wave function is defined as a curvilinear integral the velocity v_F.”

However, this text does not admit that the term in Eq. (10) is unphysical, nor does it adequately explain or give evidence for how this could possibly describe a physical dynamic process. I understand that you’re saying v_F is related to the phase of the pseudo-wavefunction since, essentially, you are treating the fermions a bit like bosons. But how is it justified to treat the dynamics of a many-fermion cloud with a single pseudo-wavefunction phase in this way?

  • validity: ok
  • significance: high
  • originality: good
  • clarity: good
  • formatting: excellent
  • grammar: good

Author:  Mariusz Gajda  on 2019-05-17  [id 518]

(in reply to Report 1 on 2019-05-02)

Evidently, we have a problem to understand the referees worry:
“But how is it justified to treat the dynamics of a many-fermion cloud with a single pseudo-wavefunction phase in this way?”

We believe that we explained in our numerous responses to the referees that Eqs.(10) are as physical as the hydrodynamic equations they originate in. Both formalisms are connected by a rigorous mathematical transformation only.

One can argue with the physical assumptions leading the hydrodynamic equations, but cannot argue with mathematical theorems.

To help the referee to understand our reasoning we want to stress again that we use a mean-field description based on the effective ONE-PARTICLE FORMALISM in the hydrodynamic form (see appendix B in arXiv:1808.04793 to get to know details on derivation of hydrodynamic equations). At the mean field level, i.e. at an effective one-particle description, a main difference between bosons and fermions is the presence of the fermionic quantum pressure, the gradient correction and A VERY STRONG CONSTRAINT ON THE VELOCITY FIELD OF FERMIONS. The pseudo-wavefucntion formalism can be transformed into hydrodynamics equations for fermions only if the velocity field is irrotational. This constraint excludes quantized vortices, for instance.

Similarity between description of condensed bosons and ultracold fermions (Eqs. (10)) is very misleading. Indeed, both equations can be classified as nonlinear Schroedinger equations. However, the similarities are stopped at this point. Let us give an example. After imprinting a phase step on a 1D BEC, a dark soliton is formed. For the uniform system, the analytical solution (Zakharov solution) for dark solitons is known. If the same phase-step is imprinted on the fermionic pseudo-wavefunction the response of the system is qualitatively different (see Phys. Rev. A 66, 023612 (2002)). After the phase imprinting, two quasisolitons, the bright and the dark ones, propagating in opposite directions are generated. Such a result is supported by the atomic-orbital calculations as well (see above mentioned paper).

Anonymous on 2019-05-22  [id 524]

(in reply to Mariusz Gajda on 2019-05-17 [id 518])
Category:
remark

I thank the referees for their latest comments on my report. I am now satisfied by their response in the sense that quantifying the physical validity of the dynamics of this theory can be left for future work.

On the one hand, I suspect that the violent dynamics of an expanding droplet may be as problematic as the interference of two separated clouds, e.g. as shown in Girardeau et al. PRL 84, 5239 (2000). On the other hand, I think that the post-instability dynamics is not really the point of the present paper, and rather it is the question of stability itself which is more important. The authors' new results (where the trap is very rapidly switched off) may make matters even worse, but still this may not be terrible in the present regime since the number of fermions is much less than the number of bosons (i.e. the expansion dynamics may be dominated by the borons).

In any case, I do not wish to delay this publication any further and I'm happy to recommend publication of the manuscript in its current form. My overall impression is that this paper represents an important first step for a very interesting system.

---

## Round 5 · Referee Report · Anonymous · 2019-5-7

Strengths

1. The work is very timely. Currently, self-bound quantum droplets represent
one the most exciting topics in the field of ultracold atoms.

2. The results are interesting and they could inspire new experiments, and also further theoretical investigation.

Weaknesses

None

Report

I'm fully satisfied with the authors' reply and with the new version of the manuscript. I am pleased to recommend it for publication.

Requested changes

None

---

## Round 5 · Referee Report · Anonymous · 2019-5-21

Strengths

1. The work is still very topical.

2. The calculations are generally well justified, with methods being compared.

Weaknesses

The discussion of three body loss is not accurate and the significance needs to be emphasised.

Report

I thank the authors again for their resubmission. I comment on the two issues from my previous report:

1. Dynamics with one Fermi wavefunction

The authors state 'We want to add that even the simplistic hydrodynamic approach is quite demanding and time of computation takes several weeks. In order to make a required comparison we decided to show results for shorter than originally presented time of opening and subsequent dynamics. In fig. 4, being a new version of the old figure 5, we show results for a total time of evolution being about 170 times shorter than in the previous one.'

For the bosons considered in Table 1, this is now turning off the trap in $50\:\mu\mathrm{s}$ to $1\:\mathrm{ms}$, so the expansion is very quickly ballistic, which doesn't provide a useful comparison of the two methods. However, I accept that this can be left for future work.

2.Three body loss

I thank the authors for rewording the section on three-body loss. I believe that what has been done is clearer. However the approximate nature of the calculation is now highlighted.

(a) The loss rate is found by extrapolation beyond the axis of Fig. 4(b) of Ref. [29] ($a_{BF}^4$ is approximate).

(b) The loss rate of Ref. [29] is for a lower density system: $5\times10^{13}\:\mathrm{cm}^{-3}$ is mentioned in [29]. The recombination parameter $K_3$ is more universal than the loss rate, and $K_3$ should be found using the density in [29] not the higher density of the current manuscript, i.e. rather than calculate $K_3 = 48/(4\times10^{14})^2 = 3\times10^{-28}\:\mathrm{cm}^6/\mathrm{s}$ as the authors do, the recombination parameter should be $K_3 = 48/(5\times10^{13})^2 = 2\times10^{-26}\:\mathrm{cm}^6/\mathrm{s}$, and when this is applied to the densities in the current manuscript, the loss rate is $2\times10^{-26}\times (4\times10^{14})^2 = 3000/\mathrm{s}$, i.e. the lifetime is two orders of magnitude lower.

(c) There should be a different loss rate for Bose-Bose-Fermi and three body Bose processes. Given this, and the approximate nature of finding $K_3$ highlighted in the previous two points, giving the results of a calculation [solving Eq. (10)] is pointless and should be removed, as requested in my previous report (I thank the authors for removing the figure).

(d) 'The loss rate exceeds the thermalization rate at $a_{BF}=-520a_0$, above which the system no longer reaches thermal equilibrium' [29] to which the authors respond 'Our calculations assume a zero temperature case and no heating due to the atom loss. For nonzero temperature, however, in particular when the loss rate exceeds the thermalization rate a dynamical nonequilibrium approach is required which is highly nontrivial and is beyond the scope of the present study.' I accept that a dynamical non-equilibrium calculation may be too onerous. However, this isn't just the issue of heating due to atom loss, but whether the atoms stay long enough for the droplet to form at all. The whole experimental feasibility of the creation of Bose-Fermi droplets is called into question by the findings of [29]. This is useful information in itself, but needs to be highlighted. The other species in Table 1 seem to be worse ($\eta_c > 2.8$), and reducing $a_B$ results in higher density and high loss, again from Table 1: $n_B = 7.16\times10^{-4}/a_B^3$.

Requested changes

1. Remove the spurious calculation of lifetimes based on Eq. (10).
2. Reduce the stated lifetime based on my point 2(b) above.
3. Reword the discussion of the implication of the thermalization rate being slower than the loss rate, so unless a more favorable system can be found by experimentalists, these droplets may never form, and highlight this in the abstract and the conclusions.

  • validity: good
  • significance: good
  • originality: good
  • clarity: good
  • formatting: good
  • grammar: good

Author:  Miroslaw Brewczyk  on 2019-05-28  [id 529]

(in reply to Report 3 on 2019-05-21)

The referee pointed out a very important issue. We want to stress that the pessimistic estimation presented by the referee is based on extrapolation of experimental data and assumptions about system densities. We still hope that a bit more optimistic (from the point of view of Bose-Fermi droplets lifetime) scenario is possible.

We suggest some modification of the abstract and conclusions. The major improvement however, is the extensive discussion of losses as described below. We hope that the referee 3 will be satisfied with the modifications.

  1. We want to modify the abstract:

We study the stability of a zero temperature mixture of attractively interacting degenerate bosons and spin-polarized fermions in the absence of confinement. We demonstrate that higher order corrections to the standard mean-field energy can lead to a formation of Bose-Fermi liquid droplets -- self-bound systems in three-dimensional space. The stability analysis of the homogeneous case is supported by numerical simulations of finite systems by explicit inclusion of surface effects. We discuss the experimental feasibility of formation of quantum droplets and indicate the main obstacle -- inelastic three-body collisions.

  1. We want to replace the discussion on losses by the text below:

The main obstacle, jeopardizing the above scenario of droplet formation is atomic loss, mainly due to three-body inelastic collisions, not included in our calculations. A crude estimation of losses can be based on the measured loss rate $\Gamma$, of Cesium atoms from a Bose-Einstein condensate immersed in a large cloud of degenerate Fermi Lithium atoms as observed in Ref. \cite{Chin17} ([29]).

To get a life time for a droplet of density $n_B=4 \times 10^{14}$cm$^{-3}$ corresponding to $a_{BB}/ a_0 =250$ and $|a_{BF}|/ a_{BB} = 3.6$, we have to extrapolate the data presented in Fig. 4b of \cite{Chin17} assuming $a_{BF}^4$ scaling of the recombination rate $K_3$. Consistently, the loss rate scales as $\Gamma \sim a_{BF}^4 n_B^2$. Assuming in addition a constant condensate density (equal to $5 \times 10^{13}$cm$^{-3}$) independent of $a_{BF}$ for all data in Fig 4b of \cite{Chin17} the estimated loss rate of atoms from the droplet is $\Gamma = 3000$s$^{-1}$. Corresponding lifetime is extremely short $\tau =0.3$ms. The estimation is very pessimistic and shows that Bose-Fermi droplets seem to be not feasible in present experiments. For $|a_{BF}|/ a_{BB} = 2.8$, i.e. at the edge of existence of the droplet, the lifetime gets longer and reaches less pessimistic value of about $\tau =1$ms.

However, the above estimation is not conclusive. Scaling of the three body recombination rate $K_3$ with $a_{BF}$ is actually more complex – the $a_{BF}^4$ behavior is modified by a factor which is an oscillating function of the scattering length \cite{PRA030703}. Simple extrapolation of data presented in \cite{Chin17} might be not precise.

\bibitem{ PRA030703} Phys. Rev. A 73, 030703 (2006)

Moreover, the estimated value of the lifetime is based on a particular interpretation of a stability of the system for a given densities of species in the regime of relatively large interspecies attraction $|a_{BF}|/ a_0 >520$ where a mean-field analysis predicts a collapse \cite{Chin17}. The stability observed in \cite{Chin17} is attributed to a dynamical equilibrium between losses of Li atoms trapped by a Cesium condensate and their supply from the Lithium vapor surrounding the Li-Cs system. Counter intuitively, the fast loss, mainly at the center of the cloud, stabilizes the system by preventing densities of both species to grow. A crucial assumption that densities of Cs atoms do not change with $a_{BF}$ is not confirmed by any data shown in \cite{Chin17}.

The experiment of C. Chin’s group \cite{Chin17} shows that a loss rate exceeds a thermalization rate in the region where the mean-field considerations predict a collapse ($|a_{BF}|/a_{BB} = 600, 700$ in Fig. 4b of \cite{Chin17}). We want to speculate that even in such a dynamical situation a formation of a droplet might be still possible if both species densities adjust to the ‘droplet values’ at dynamical equilibrium. The situation could resemble to some extent a polariton condensate – a lifetime of its components is much smaller than the coherence time of the system which is at dynamical equilibrium.

Therefore, an alternative origin of observed stability can be attributed to a repulsion of atoms due to beyond mean-field effects leading to formation of droplets of densities depending on $a_{BF}$. In such a case interpretation of Fig. 4b of \cite{Chin17} must be different, and accounting for densities depending on scattering length will significantly influence $K_3$ scaling with $a_{BF}$. Extrapolation of loss rate is very difficult then.

To support this point we would like to note that Fig. 2b of \cite{Chin17} already shows an elongated falling object, living for at least by $2.5\,$ms, whose existence cannot be explained by the dynamical model proposed by the authors since the lack of overlap with the fermionic background cloud which was pushed upwards. We performed numerical simulations for Cs-Li mixture for parameters as studied in \cite{Chin17}. Our calculations show that already for $a_{BF}/a_B = -2.8$ an elongated droplet is formed in the trap with the bosonic density of $n_B=4\times 10^{14}$cm$^{-3}$ which after removal of the trapping potential survives and oscillates. Our simulations are in agreement with results shown in Fig. 2b, supporting Bose-Fermi droplets scenario.

The above discussion is highly speculative. Definitely much more experimental and theoretical work is needed to find out what will be the fate of the Bose-Fermi droplets discussed here. The pessimistic estimation of droplet’s life time presented above has to be treated as a serious warning but no definite conclusion about a value of loss rates in the droplet regime can be drawn on the basis of \cite{Chin17}.

  1. We want to rephrase the first paragraph of conclusions.

We suggest:

The analysis of stability of a mixture of ultracold Bose-Fermi atoms presented here indicates that stable liquid self-bound droplets can be spontaneously formed when interspecies attraction is appropriately tuned. Droplets are stabilized by the higher order term in the Bose-Fermi coupling. We predict the values of interaction strengths as well as atomic densities corresponding to droplets of three different mixtures, suitable for experimental realization, $^{41}$K-$^{40}$K, $^{87}$Rb-$^{40}$K, and $^{133}$Cs-$^{6}$Li.

We demonstrate by time dependent calculations that a Bose-Fermi droplet should be achievable by preparing the mixture of bosonic and fermionic atoms in a trap and then by slowly removing the confinement. The main obstacle on a way to form the droplets are three-body losses. The droplets are formed in a regime where inelastic collisions are not negligible. Unfortunately, the existing experimental data do not allow to determine unambiguously the lifetimes of the droplets. The crude estimation based on extrapolation of the loss rate is very pessimistic, showing that Bose-Fermi droplets are illusive objects. The second scenario, assuming beyond mean-field effects play essential role, is much more optimistic. Moreover, we would like to note that low dimensional Bose-Fermi droplets (arXiv:1808.04793) are free of three-body loss related troubles and should be experimentally feasible soon.

---

## Round 5 · Author Response

Dear Editor,

We want to thank you for handling our manuscript.

We are very grateful to the referees who found our work interesting and physically sound.

The present version of our manuscript is modified according to the requests of the Referees.

In particular we performed dynamical simulations showing behavior of the mixture released from the trap for different parameters. The calculations, seemingly simple were quite demanding so we were not able to respond shortly.

Below we give detailed answers to all requests of the three referees. We hope that the present version of our manuscript will be accepted for publication.

Sincerely Yours,

Debraj Rakshit Tomasz Karpiuk Miroslaw Brewczyk Mariusz Gajda

Answer to the editor’s and referees’ requests and a list of changes

Editor’s requests: 1. Please carefully consider the changes recommended in each of the three referee reports. 2. Please discuss S K Adhikari 2018 Laser Phys. Lett. 15 095501 in your resubmission.

Our response: Ad. 1. Bellow we carefully answer to the all requests and recommendations of every of three referees. Ad. 2 As suggested by the editor we added the following paragraph commenting on the recent work on Bose-Fermi droplets by S.K. Adhikari: In a recently published work, Ref. \cite{Adhikari18}, a repulsive short-range three-bosons interactions are added to stabilize the Bose-Fermi mixture. This mechanism was previously suggested in \cite{Blakie16} to stabilize a dipolar condensate. Unfortunately the mechanism is rather difficult to implement because large three-body elastic collisions are typically accompanied by large three-body losses. In addition, in \cite{Adhikari18} it is assumed that fermions are in a fully-paired superfluid state, what in fact makes the system similar to a Bose-Bose rather than to a Bose-Fermi mixture. And finally, the Ref. \cite{Adhikari18} shows that the droplets may consist of bosonic and fermionic atoms in almost equal ratio in contrary to our results indicating a significant domination of the Bose component.

Referee 1 requests the following changes:

The authors should either admit that the term in Eqs. (10) which involves the ‘phase’ of the single fermion pseudo-wave function (which they use to describe the non-adiabatic dynamics of many fermions) is unphysical, or they should explain what physical process it actually represents, and give evidence for this.

In order to meet the referee's demand we added the paragraph:

We would like to emphasize that the fermionic pseudo-wave function has no direct physical meaning. Only the quantities which are the square of modulus of $\psi_F({\bf r},t)$ and the gradient of its phase can be interpreted as physical quantities. The Madelung transformation itself is supported by the Stokes' theorem. Provided that in a given region the condition $\nabla \times \vec v_F=0$ is fulfilled, then the phase of the pseudo-wave function is defined as a curvilinear integral of the velocity $\vec v_F$.

The Referee 2 has the following concerns:

  1. Dynamics with one Fermi wave function

I am still concerned that the solid curve in Fig. 3(left) may not be approximately correct, as it uses a single wave function with a single phase for an expanding gas. The behaviour of the solid line Fig. 3(left) cannot be easily compared to the black curve in Fig. 5(right) as the quench, timescales and number of atoms are quite different. In Fig. 5(right), please include dynamical results from the hydrodynamical approach with the same parameters as for the atomic-orbital approach. This will allow easy comparison, as we have for equilibrium results in Fig. 5(left).

We performed additional calculations. We want to add that even the simplistic hydrodynamic approach is quite demanding and time of computation takes several weeks. In order to make a required comparison we decided to show results for shorter than originally presented time of opening and subsequent dynamics. In fig. 4, being a new version of the old figure 5, we show results for a total time of evolution being about 170 times shorter than in the previous one. We substituted the paragraph commenting the numerical results by the following one:

On the other hand, in the right frame of Fig. 4 we compare the dynamical properties of the Bose-Fermi mixture for different values of the mutual scattering length aBF, obtained within the atomic-orbital and hydrodynamic approaches. Here, the trapping potential is removed in 1ms (marked by a vertical line) as in the case of Fig. 3. Similarly to the previous analysis, only for large enough |aBF|/aB (>2.8) a droplet is formed, otherwise we observe an expansion of both atomic clouds. Note however that for |aBF|/aB close to the critical value, there appears a small discrepancy between both descriptions. But it only means that the critical values of |aBF|/aB found within the atomic-orbital and hydrodynamic analyses are slightly different. This is because of relatively large contribution of the surface terms to the total energy for such small systems. These terms are treated on a different footing in both compared methods. Away of the critical value of |aBF|/aB both approaches match perfectly.

  1. Three body loss

The calculation for Fig. 4 is too approximate to be useful. The quench of scattering lengths and trap in an experiment will lead to oscillations which will affect loss, there being a trade-off between a fast quench (desirable due to short lifetimes) but stronger oscillations. The experiment will also have noise which has not been added in the calculations. I suggest removing Fig. 4 and coming up with a broad estimate based on the rate coefficient. Also the paragraph on three body loss is confusing. Please state clearly what you get from where. You are considering a case of aB=250a0 and aBF=−3.6aB=−900a0? The bosonic density of nB=0.0009/a3B=3×1014cm−3 is taken from your calculations (the manuscript says 'From the rate equation')? I see you have used K3=Γ/n2B, but where did your Γ=10/s come from? Please do not state K3 to three significant figures. Please state clearly how you get Γ=50/s from Fig. 4(b) of [29] including how you allow for your values of aB and aBF and for your increased density.

We have added the following discussion:

Finally, we address the issue of a life-time of a Bose-Fermi droplet due to only three-body recombination processes, neglecting all the other sources of atomic loss. A crude estimation of losses could be done based on the loss dynamics of Cs condensate atoms immersed in Li degenerate fermions as observed in Ref. [29], see Fig. 4. For example, for aB= 250 a0 and the range of |aBF|/aB (2.8,3.6), the loss rate Γ=(1/NB) dNB/dt can be extracted from Fig. 4b of [29] assuming the expected Γ ~ aBF^4 dependence. The resulting life-time is in the range 15 ms-30 ms. The finer analysis could rely on solving Eqs. (10) with additional terms, representing losses resulting from Li-Cs-Cs collisions. The right-hand side of Eqs. (10) is then modified by adding (-iK3/2 nBnF) and (-iK3/2 nB^2) terms for bosonic and fermionic components, respectively. Here, K3 is the rate coefficient which is estimated from the rate equation (1/NB) dNB/dt = -K3. For example, for aBF/aB = -3.6 and aB= 250 a0 one has Γ = (1/NB) dNB/dt = 50/s (from Fig. 4b in [29]) and for nB = 4 \times 10^{14}cm^{-3} we find K3 = 3 \times 10^{-28} cm^{6}/s. Now, solving Eqs. (10) gives the life-time of the droplet to be about 45 ms.

Please also discuss how 'The loss rate exceeds the thermalization rate at aBF=−520a0, above which the system no longer reaches thermal equilibrium' [29] relates to your system.

The case of nonequilibrium dynamics requires different approach. We added the following comment:

Our calculations assume a zero temperature case and no heating due to the atom loss. For nonzero temperature, however, in particular when the loss rate exceeds the thermalization rate a dynamical nonequilibrium approach is required which is highly nontrivial and is beyond the scope of the present study.

Referee 3 requests the following changes:

1) The author state in the abstract that "Bose-Fermi liquid droplets -- self-bound incompressible systems" are formed, but then the (in)compressibility of the system is never discussed. The authors should clarify this point, are these systems really incompressible?

We agree with the referee. Compressibility of the BF droplets requires detailed studies. We modified abstract and removed expressions suggesting incompressible character of droplets.

2) The choice of the time unit -- ħ/(mBaB2) -- is very uncommon. Why not using milliseconds or something more immediate to read? I would recommend to change it or explain clearly in the text the reason of this choice.

We decided to express time in units related to the energy related to the scattering length a_B^2 rather than milliseconds. To use milliseconds we should chose some particular value of the Bose-Bose s-wave scattering length. For the exemplary case of Cs-Li mixture used here there are two different regimes of parameters used by the Cheng Chin's group, namely aB=250 (Ref. [29]) or 4 (Ref. [26]) Bohr radii. These two values lead to the very different time scales. This is why we decided not to translate the dimensionless values of time to milliseconds. However, in Figs. 3 and 4b (new version) we included a vertical solid lines which show the moment of time equal to 1ms, so one could easily read the typical time scales (for aB=250 a0).'

3) The content of Fig. 4 does not justify the need of a figure. Its meaning could be easily explained by adding a text line.

We removed Fig. 4 and added the explanation instead.

4) Which is the physical origin of the oscillations in Fig. 4right? Have the authors checked that this is not just a numerical effect? Please explain in the text.

The effect is physical. These oscillations are caused by the mismatch of the rate with which the trap is open and the rate of mixture's expansion. The trap is opened very slowly while expansion is very fast, so expanding atoms are back-reflected by the slowly softened trap. This scenario is repeated then.

The Referee 2 asked us to perform the additional calculations allowing for a direct comparison of the expansion obtained on the ground of the pseudo-wave function formalism to the one given by the orbital method. The calculations based on the pseudo-wave function, are quite demanding and require weeks of computations (the advantages of the pseudo-wave function approach becomes evident for large systems). To save time we decided to show results for a fast opening of the trap. Therefore in Fig. 4 (which substitutes for the old figure 5) the oscillations of the radius of the cloud are not present. Opening of the trap is so fast that expanding cloud is not back-reflected from the “trap walls”.

5) The English may need some revision. Please check carefully the (missing) articles. We made our best to correct English.

---

## Round 5 · List of Changes

List of changes:

1. The paper by S K Adhikari 2018 Laser Phys. Lett. 15 095501 is discussed.

2. We discuss the mathematical foundations of Eqs. (10).

3. We compare the dynamics of the Bose-Fermi mixture obtained by the hydrodynamic and atomic-orbital approaches, Fig. 4b (which replaced Fig. 5b from the previous version).

4. We removed Fig. 4 (old version) and discuss losses based on experimentally measured loss dynamics and on the rate coefficient, K3, deduced from the experiment.

5. We comment the referee's question about experimentally measured loss rate at aBF=−520a0 and its relation to the thermalization rate (Ref. [29]).

6. We removed the word 'incompressible' from the abstract.

7. In Figs. 3 and 4b there are vertical solid lines now, which show the moment of time equal to 1ms to allow the reader to immediately read the time scales.

8. We improved English.

---

## Round 6 · Author Response

Dear Editor,

We are resubmitting a new version of our paper modified by including a discussion of three-body losses as required by the third referee.

Sincerely Yours,
the authors

---

## Round 6 · List of Changes

1. We modified the last sentence of the abstract.

2. We replaced a single paragraph on the three-body losses by the extensive discussion of losses in the context of Ref. [29].

3. We replaced the first paragraph of conclusions by two new ones emphasizing now the significance of losses for the observation of the Bose-Fermi droplets.

---

## Editorial Decision

published